# A robust intrusion detection system based on a shallow learning model and feature extraction techniques

**Chadia E. L. Asry[1], Ibtissam Benchaji[1], Samira Douzi [ID][2]\*, Bouabid E. L. Ouahidi[1]**

**1** IPSS Laboratory, Faculty of Sciences, Mohammed V University in Rabat, Rabat, Morocco, **2** Faculty of Medicine and Pharmacy, Mohammed V University in Rabat, Rabat, Morocco

\* s.douzi@um5r.ac.ma

**Data Availability Statement:** The data are held in a public repository: https://research.unsw.edu.au/projects/unsw-nb15-dataset. https://www.unb.ca/cic/datasets/nsl.html.

## Abstract

The escalating prevalence of cybersecurity risks calls for a focused strategy in order to attain efficient resolutions. This study introduces a detection model that employs a tailored methodology integrating feature selection using SHAP values, a shallow learning algorithm called PV-DM, and machine learning classifiers like XGBOOST. The efficacy of our suggested methodology is highlighted by employing the NSL-KDD and UNSW-NB15 datasets. Our approach in the NSL-KDD dataset exhibits exceptional performance, with an accuracy of 98.92%, precision of 98.92%, recall of 95.44%, and an F1-score of 96.77%. Notably, this performance is achieved by utilizing only four characteristics, indicating the efficiency of our approach. The proposed methodology achieves an accuracy of 82.86%, precision of 84.07%, recall of 77.70%, and an F1-score of 80.20% in the UNSW-NB15 dataset, using only six features. Our research findings provide substantial evidence of the enhanced performance of the proposed model compared to a traditional deep-learning model across all performance metrics.

## 1. Introduction

With the rise in frequency and complexity of cyber-attacks targeting networks and data [1], the predominant approach to mitigating these attacks revolves around the utilization of classifiers. In general, these methodologies ascertain the presence of attacks by categorizing network packets as either benign or exhibiting anomalies. Certain attack detection systems also categorize intrusion alerts according to the level of severity associated with the intended harm [2]. Nevertheless, all these methodologies are fundamentally responsive in nature, as they primarily concentrate on retaliatory measures. Ongoing efforts are being made to develop systems with the capability to accurately forecast an impending attack.

In the realm of cybersecurity, Intrusion Detection Systems (IDSs) are commonly categorized into two distinct types: signature-based IDSs and anomaly-based IDSs. Signature-based intrusion detection systems (IDSs) employ a predetermined database of attack signatures to identify events and network traffic that exhibit characteristics consistent with known attack

**Funding:** The author(s) received no specific funding for this work.

**Competing interests:** The authors have declared that no competing interests exist.

patterns. This approach is employed to mitigate privacy concerns and counteract potential security threats [3]. Nevertheless, their capacity to detect unforeseen patterns and indicators in novel attacks is constrained [4]. Conversely, anomaly-based intrusion detection systems (IDSs) endeavor to comprehend typical patterns of behaviours and categorize any deviations as anomalies or intrusions. Nevertheless, these systems tend to generate a considerable number of inaccurate results, thereby greatly diminishing their efficacy.

Several intrusion detection systems (IDS) methodologies have been documented in scholarly works, utilizing diverse machine learning models to autonomously differentiate between regular and anomalous occurrences within systems and networks [5]. Machine learning is an algorithmic approach in which a computational system acquires the ability to process data by discerning patterns and rules from a set of training examples. The method described can be categorized into two main groups: shallow learning and deep learning [6]. Shallow learning commonly utilizes a limited number of layers for data processing, whereas deep learning involves the utilization of numerous consecutive layers [6]. Deep learning has demonstrated significant accomplishments in a range of fields, including visual comprehension, natural language processing, computer vision, and Automatic Modulation Classification [6–12].

Shallow learning, as a widely recognized technique, has been extensively employed in the field of cybersecurity for the purpose of attack detection. Numerous studies have employed the random forest algorithm [13], support vector machine (SVM) [14], and various other algorithms to classify intrusions from traffic data.

Machine learning (ML) driven intrusion detection systems (IDS) possess enhanced capabilities in the identification of novel patterns or behaviours. The integration of natural language processing (NLP) can enhance the detection accuracy of these intrusion detection systems (IDSs). NLP-based detection mechanisms offer advantages over techniques that rely on specific attack methods.

The subsequent content outlines the key contributions of this research article:

- This research presents a novel intrusion detection system (IDS) that incorporates a feature selection process. The IDS leverages the PV-DM shallow learning model and employs multiple machine learning models for the purpose of attack classification.

- Two approaches for feature reduction, namely Mutual Information (MI) and SHAP values, are utilized to minimize unnecessary features that could potentially have a detrimental effect on classification performance.

- The PV-DM methodology is employed to convert intrusion data into feature vectors, which are subsequently employed for training machine learning models.

- A sequence of experiments was undertaken to evaluate the efficacy of several machine learning methodologies, with the objective of determining the most effective classifier and feature selection model.

- The work incorporates a comparative analysis between the suggested model and LSTM, a prevalent deep learning model utilized in intrusion detection systems.

- The evaluation demonstrates that the proposed approach exhibits superior performance compared to the LSTM model. This study represents the initial endeavor to evaluate the performance of LSTM and PV-DM models within the domain of intrusion detection.

The present paper is organized into subsequent sections: The following section delves into the literature that is relevant to the topic at hand. In section 3, we outline the different theoretical frameworks and concepts that informed our research. Section 4 offers a thorough

exposition of the proposed methodology, encompassing implementation particulars and empirical findings. In conclusion, Section 5 serves as the final part of the article.

## 2. Previous works

A multitude of scholarly investigations have been conducted to examine the application of deep learning methodologies in the development of robust Intrusion Detection Systems (IDSs) capable of safeguarding against diverse forms of cyber threats. Numerous scholarly articles have been published on the subject matter, encompassing various strategies and methodologies that have proven effective in developing resilient Intrusion Detection Systems (IDSs).

Mamoru [15] utilized a generic detection method known as DOC2VEC to conduct a timeline analysis and cross-dataset validation on the D3M and MTA datasets. These datasets consisted of captured Exploit Kit (EK) traffic. The outcomes demonstrated a high level of effectiveness, as evidenced by a detection accuracy of 0.98 on the timeline analysis dataset and 0.97 on the other dataset, as measured by the F-measure. This study showcased the efficacy of the methodology in accurately identifying and detecting current exploit kit (EK) traffic. The study conducted by San [16] employed skip-gram modelling, which is a variant of the word2-vec technique, for the purpose of conducting testing. The intrusion detection algorithm demonstrated notable performance metrics, including a precision level of 99.20%, a recall level of 82.07%, and an accuracy level of 91.02%. Additionally, it effectively minimized false positives, with a low rate of 0.61%. The findings exhibited a notable improvement in comparison to the outcomes achieved through contemporary methodologies. The validation of the model was conducted using the UNSW-NB15 dataset. The authors of the study presented in [17] introduced a system that combines two deep learning models, namely Long Short-Term Memory (LSTM) and Convolutional Neural Network (CNN). The CNN-LSTM model was assessed on the KDD99 dataset, yielding a precision rate of 99.78%. The study primarily concentrated on a singular performance parameter and employed a sole dataset to assess the proposed model. The research conducted in reference [18] presented a novel intrusion detection system based on artificial intelligence (AI), specifically employing a deep neural network (DNN). The evaluation of the system was conducted on the KDD Cup 99 dataset, employing a Deep Neural Network architecture consisting of four hidden layers. The classification type employed in the study was binary, distinguishing between normal and attack instances. The system demonstrated a commendable accuracy rate of 99.08%. The authors in reference [19] presented a novel academic contribution, wherein they proposed a Deep Learning framework that integrates Auto-Encoders and Deep Belief Networks (DBN) to enhance the effectiveness of intrusion detection. The utilization of an Auto-Encoder facilitated the reduction of data dimensionality and the identification of its salient features, whereas the Deep Belief Network (DBN) was tasked with the detection of potentially malicious code. The model under consideration was assessed using the KDD Cup 99 dataset and was compared to a solitary Deep Belief Network (DBN). The findings indicate that the newly developed hybrid system exhibits a statistically significant improvement in both accuracy and efficiency. Nevertheless, the authors did not furnish an elaborate rationale for their decision to amalgamate Deep Belief Networks (DBN) with Auto-Encoder within their model. In their study, Kim et al. [20] conducted a comparative analysis of various intrusion detection techniques using the KDD99 dataset. The researchers discovered that the LSTM-RNN network exhibited superior performance compared to various other methods, such as the Generalized Regression Neural Network (GRNN), Product-based Neural Network (PNN), k-Nearest Neighbors (KNN), Support Vector Machine (SVM), Bayesian, and other approaches. The authors in reference [21] introduced a novel intrusion detection system (IDS) based on deep learning, specifically integrating

Convolutional Neural Networks (CNN) and Long Short-Term Memory Networks (LSTM). The DL-IDS was evaluated on the CICIDS2017 dataset using a multiclassification approach, resulting in an impressive overall accuracy of 98.67 percent. Furthermore, they were able to achieve an accuracy rate of over 99.50 percent for every type of attack. Yakubu et al. [22] introduced a bidirectional Long-Short-Term Memory (BiDLSTM) approach for intrusion detection, as documented in their study. The performance of this system was assessed using the NSL-KDD dataset. The experimental findings demonstrated that the BiDLSTM model exhibited superior performance compared to other contemporary models, such as LSTM, in terms of accuracy, precision, recall, and F-score metrics for the detection of U2R and R2L attacks. Shen et al. [23] introduced a hybrid methodology that integrated the extreme learning machine (ELM) and the Bat Algorithm (BA) in order to enhance the accuracy of intrusion detection by optimizing feature selection. The Extreme Learning Machine (ELM) was utilized as the principal classifier, while the Boosting Algorithm (BA) was employed to eliminate redundant features, thereby reducing the dimensionality of the dataset. The method proposed by the researchers was assessed using three distinct datasets, namely KDD99, NSL-KDD, and Kyoto. The evaluation resulted in an overall accuracy of 98.94%. Dong [24] introduced a novel intrusion detection model named AE AlexNet, which utilizes the Auto-Encoder AlexNet neural network architecture. When the model was assessed using the KDD99 dataset, it demonstrated a level of accuracy amounting to 94.32 percent. The authors in reference [25] introduced DL-MAFID, an intrusion detection model based on a multi-agent system, which incorporates an auto-encoder for the purpose of dimensionality reduction. The proposed model has been specifically developed to address the complexities associated with multi-class intrusion detection tasks. Its performance is assessed using the KDDCup'99 dataset. In addition to utilizing the auto-encoder, the dataset's five categories are classified using shallow classifiers such as MLP and KNN. The experimental findings demonstrate that DL-MAFID can attain a detection accuracy of 99.95 percent, while concurrently diminishing the duration required for detection. In their study, the authors presented three distinct intrusion detection systems that leverage deep learning techniques, specifically Artificial Neural Networks (ANN), Deep Neural Networks (DNN), and Recurrent Neural Networks (RNN) [26]. The evaluation of the proposed methodologies is conducted using the UNSW-NB15 dataset, encompassing both binary and multiclass classification tasks. The authors in [27] propose the introduction of BFFA, which is a binary adaptation of the Agricultural Land Fertility Algorithm (FFA). BFFA is specifically developed for the purpose of feature selection in the classification of Intrusion Detection Systems (IDS). The classifiers DT, KNN, and SVM were hybridized, integrating the suggested feature selection (FS) technique to create an efficient and reliable intrusion detection system (IDS). The methodology was subjected to a comparative analysis with several algorithms, namely K-Nearest Neighbours (KNN), Support Vector Machines (SVM), Decision Trees (DT), Random Forests (RF), Adaptive Boosting (ADA_BOOST), and Naive Bayes (NB). The NSL-KDD and UNSW-NB15 datasets were subjected to evaluation, revealing that the suggested technique outperformed previous classifiers in terms of Accuracy, Precision, Recall, and runtime.

In their study, the researchers provide MODHHO, a Multi-Objective Dynamic Harris Hawks Optimization technique, specifically developed for the purpose of detecting Botnets in Internet of Things (IoT) networks [28]. By utilizing the powerful HHO algorithm, MODHHO effectively detects significant features inside Internet of Things (IoT) datasets, exhibiting enhanced performance through the implementation of a multi-objective strategy. The K-Nearest Neighbor (KNN) classifier is utilized to evaluate classification errors, indicating that MODHHO has a high level of precision compared to its competitors in the identification of IoT botnets. The model that has been suggested consistently demonstrates superior performance

compared to alternative algorithms, resulting in an average improvement of 26% across all datasets and evaluation criteria.

Table 1 presents a summary of several established Intrusion Detection System (IDS) models.

Based on the literature assessment, it is evident that most of the research lacks emphasis on data feature engineering models, which play a crucial role in identifying relevant features. Furthermore, even when some studies employ a reduction technique, they tend to rely solely on a single method. Furthermore, the majority of the research assessed their proposed models by employing a solitary dataset and using binary classification. It is noteworthy that recent research investigations have placed significant emphasis on the utilization of intrusion detection systems (IDS) based on deep learning techniques. However, it is important to consider that such systems may possess inherent complexities and require substantial time investments. In contrast, practical IDS systems should prioritize user-friendliness, operational efficiency, and the ability to promptly detect anomalies. Moreover, it is evident that there has been no prior research that has integrated the PV-DM architecture with Mutual Information and

**Table 1. Overview of various well-established models in intrusion detection systems (IDS).**

| Ref | Architecture | Dataset | Result | Limitations |
|---|---|---|---|---|
| [18] | DNN | KDD Cup 99 | Accuracy = 99% | The study utilized a binary classification approach. |
| [19] | Auto-Encoders and Deep Belief Networks (DBN) | KDD Cup 99 | Accuracy = 92.10% | The authors did not include an extensive rationale for their decision to use Deep Belief Networks (DBN) with Auto-Encoder within their model. |
| | | | FPR = 1.58% | |
| | | | TPR = 92.20% | |
| [21] | Convolutional Neural Networks (CNN) and Long Short-Term Memory Networks (LSTM) | CICIDS2017 | Accuracy = 98.67% | A comparative analysis of DL-IDS in relation to other established or rival methodologies is absent from the article. An examination of the proposed model in comparison to alternative Intrusion Detection System (IDS) methods might have provided further understanding of its distinct benefits. |
| | | | F1-score = 93.32% | |
| | | | TPR = 97.21% | |
| | | | FPR = 0.47% | |
| [29] | Random Forest, Unsupervised model | KDDCUP99 | Accuracy = 96.42% | Longer execution time |
| | | | FPR = 0.98% | |
| [30] | Bidirectional Long short-term memory networks Recurrent Neural Network BLSTM-RNN | - | The results for Mirai, UDP, and DNS show a validation accuracy of 99%, 98%, and 98%, along with corresponding validation loss metrics of 0.000809, 0.125630, and 0.116453, respectively. | Longer execution time |
| [31] | RF regressor+ ID3 | CICIDS2019 | Precision = 78% | The efficacy of the proposed model is poor. |
| | | | Recall = 65% | |
| | | | F1-score = 69% | |
| [32] | CFS-BA | NSL-KDD, CIC-IDS2017, AWID | Accuracy = 99.8% | Performance is not significantly improved through the use of ensemble methods, especially in situations where there is a substantial discrepancy in the distribution of classes. |
| | | | Precision = 99.8% | |
| | | | Recall = 99.8% | |
| | | | F1-score = 99.8% | |
| | | | Accuracy = 99.9% | |
| | | | Precision = 99.9% | |
| | | | Recall = 99.9% | |
| | | | F1-score = 99.9% | |
| | | | Accuracy = 99.5% | |
| | | | Precision = 99.5% | |
| | | | Recall = 99.5% | |
| | | | F1-score = 99.5% | |

SHAP value dimension reduction methods for the purpose of binary and multiclass classification.

## 3. Materials and methods

### 3.1 Shallow learning and deep learning

**3.1.1 Shallow learning: Paragraph Vector-Distributed Memory algorithm.** A shallow learning model, sometimes referred to as shallow machine learning or shallow neural network, denotes a machine learning model characterized by a restricted number of layers or architectural complexity. Shallow learning models are commonly characterized by their limited depth, typically comprising only one or a few layers of neurons. These models are frequently employed for tasks or datasets that exhibit basic characteristics, where the associations between features and targets are relatively uncomplicated. The Paragraph Vector-Distributed Memory (PV-DM) algorithm, also known as Doc2Vec, is a commonly employed method for creating distributed representations of text segments of varying duration, such as sentences, paragraphs, or complete documents [33]. The model discussed in this paper is an adaptation of the Word2Vec model (Mikolov et al., 2013) [34], which is designed to represent the semantic meaning of words in a continuous vector space.

In the PV-DM framework, a distinct vector is assigned to each paragraph, which is denoted by a column in the matrix D. Similarly, each word is likewise assigned a unique vector, represented by a column in the matrix W (Fig 1) [35]. The paragraph vectors and word vectors are subsequently employed in conjunction to forecast the subsequent word within a provided

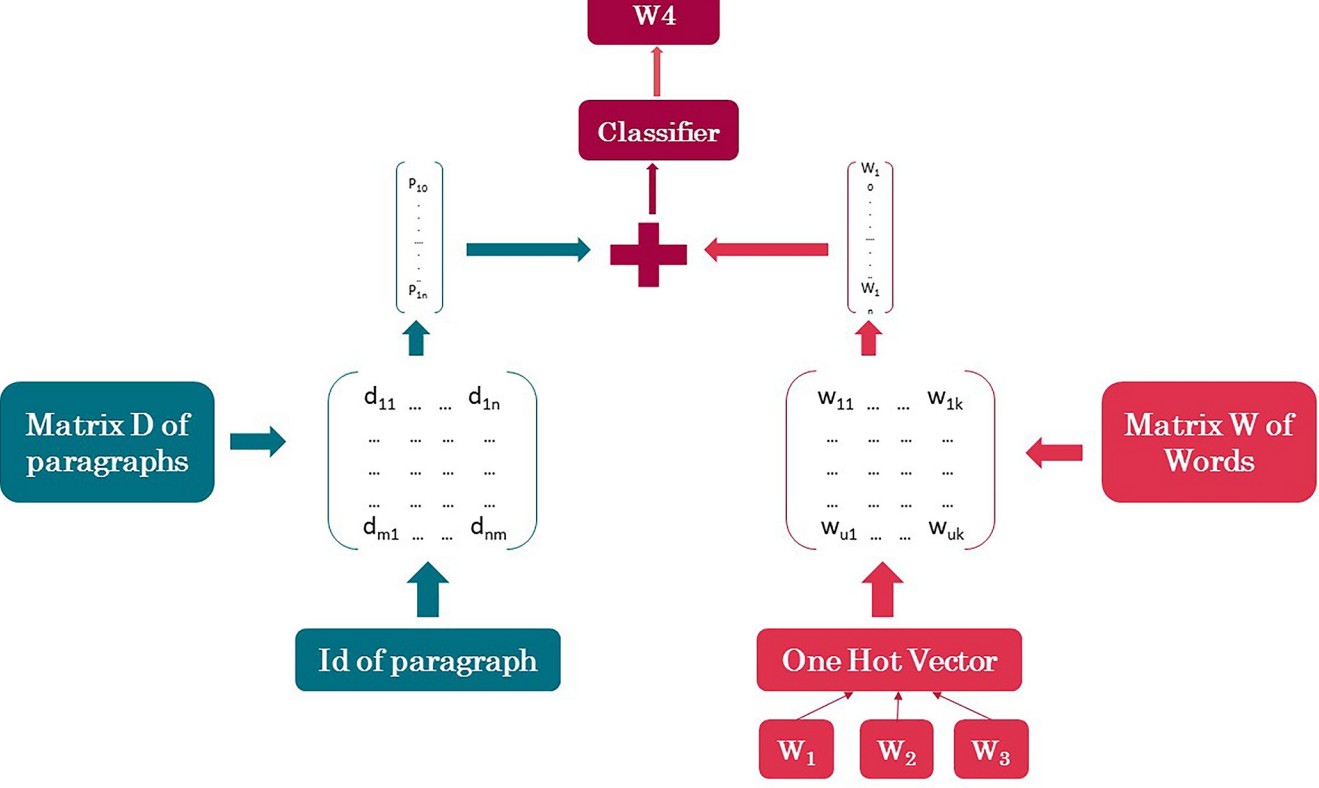

**Fig 1. A framework for learning paragraph vectors [35].**

context. The process of combining these vectors is accomplished through the use of concatenation. The following items are available.

$$p\left(W_t|W_{t-k},\ldots,W_{t+k};d,W,D\right) = \frac{e^{y_{wt}}}{\sum_i e^{y_i}} y = b + Uh\left(W_t|W_{t-k},\ldots,W_{t+k};d,W,D\right)$$

Where b and U are classifier parameters, h is derived from W and D, and d is the vector of the paragraph from which the words $w_{t-k},\ldots,w_{t+k}$ are issued.

**3.1.2 Deep learning: Long Short-Term Memory (LSTM).** The Long Short-Term Memory (LSTM) model, which was first proposed by Hochreiter and Schmidhuber in 1997, is a distinct variant of the recurrent neural network (RNN) [36]. The proposed solution tackles the difficulties associated with the issues of disappearing and exploding gradients in conventional recurrent neural network (RNN) training. This is achieved by integrating three distinct gates, namely the input gate, the output gate, and the forget gate. The utilization of these gates allows the Long Short-Term Memory (LSTM) model [37] to effectively and selectively retain or discard information over a given period. The input gate, output gate, and forget gate are represented as $i_t$, $o_t$, and $f_t$, respectively, at each time step t (Fig 2) [38].

■ The input/update gate layer is responsible for determining if new values obtained from incoming data can be allowed to pass through. This decision is made using two functions: "sigmoid" and "tanh". The "sigmoid" function assigns a value of zero to restrict modification, while a value of one permit modification. On the other hand, the "tanh" function generates a new vector of values that will be added to the cell state. The calculation is performed in the following manner:

$$i_t = \sigma\left(W_i[h_{t-1},x_t] + b_i\right) \text{ and } c_t = \tanh\left(W_c[h_{t-1},x_t] + b_c\right)$$

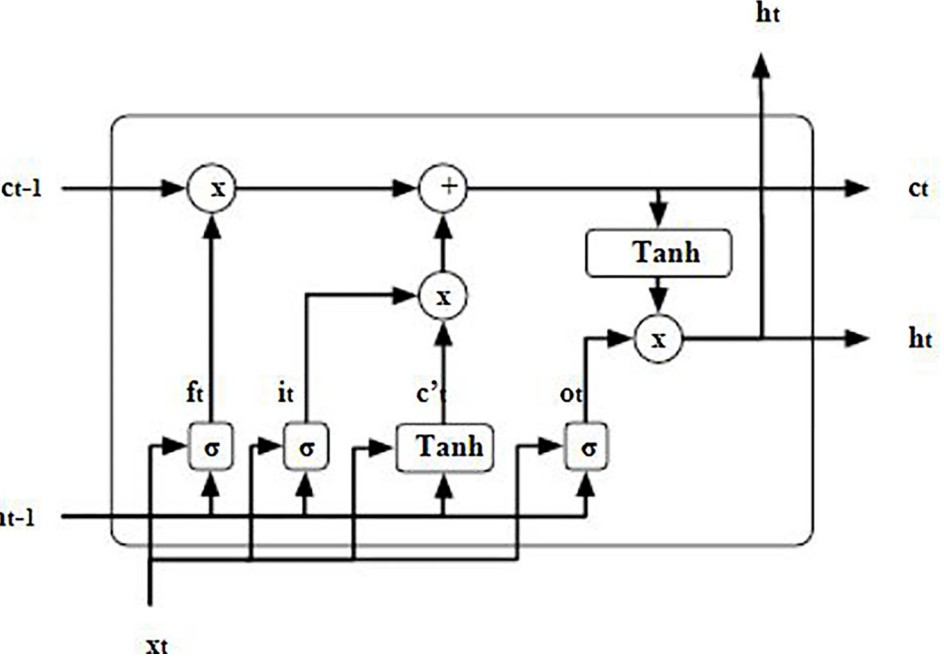

**Fig 2. Long short-term memory (LSTM) architecture.**

■ The forget gate layer utilizes a sigmoid function to determine the selective passage of data between the current input and the previous cell state output. The calculation is performed in the following manner:

$$ft = \sigma(W_{f.}[h_{t-1,}, X_t] + b_f)$$

■ The Output Layer is responsible for determining the specific information that is to be outputted from the cell state. Before using the "sigmoid" function, the output is first subjected to the "tanh" function in order to ensure that the values fall within the range of -1 to +1. The calculation is performed in the subsequent manner:

$$O_t = \sigma(Wo[h_t - 1, x_t] + b_0)h_t = O_{t*} \tanh(h_t)$$

Theoretically, the employment of a limited number of layers in shallow learning enhances the interpretability of judgments, which is crucial for interpretation. The prioritization of interpretability is highly significant, particularly within crucial domains like intrusion detection. From a practical standpoint, shallow learning models demonstrate exceptional performance in scenarios when data is scarce and necessitate a less amount of labeled data for training purposes. In contrast, deep learning methodologies may require a significant amount of annotated data to achieve optimal effectiveness and include longer execution periods due to the utilization of several layers. Moreover, shallow models exhibit reduced computing complexity, hence enhancing their suitability for deployment in resource-constrained settings. In the context of intrusion detection, the prioritization of real-time threat detection necessitates the adoption of a shallow model that provides enhanced interpretability. This approach proves to be more feasible and effective, especially when considering the limitations of deep learning in terms of data requirements and execution times.

### 3.2 Features selection

The process of feature selection is of utmost importance as it entails the selection of a subset of significant features from a given dataset [39]. This procedure presents numerous advantages, such as the reduction of computing expenses, the improvement of model efficacy, and the facilitation of model understanding for researchers.

The objective of our research was to reduce the dimensionality of the datasets by a comparative analysis of two feature selection procedures, namely Mutual Information (MI) and SHAP values. These algorithms offer techniques for evaluating the significance and relevance of individual features in relation to the target variable. Through the application of MI (Mutual Information) and SHAP (Shapley Additive Explanations) values, it becomes possible to discern the most influential features and eliminate extraneous or redundant ones. This process effectively reduces the overall dimensionality of the dataset, resulting in enhanced model efficiency and accuracy, all the while preserving the integrity of the data.

### 3.2.1 Mutual information (MI)

The effectiveness of mutual information (MI) in filter feature selection algorithms has been demonstrated in examining the predictive potential of a subset of features and their redundancy with other variables [40]. The concept of mutual information (MI) involves the quantification of the information that may be acquired by monitoring a random variable in relation to another. It entails assigning a numerical value to each feature based on this measure. In the context of statistical analysis, it is observed that when two variables are deemed to be

statistically independent, the mutual information (MI) value between them is determined to be 0. However, when the level of dependence between the variables intensifies, the MI value progressively increases. By employing mutual information (MI) as a criterion for the selection of features, it becomes possible to discern features that contain substantial information pertaining to the target variable, while simultaneously excluding redundant or unnecessary features.

The concept of the MI is defined as follows:

$$I(X; Y) = \sum_{x,y} P_{XY}(x, y) \, log \frac{P_{XY}(x, y)}{P_X(x) P_Y(y)} = E_{P_{XY}} log \frac{P_{XY}}{P_X P_Y}$$

Which X and Y are two discrete variable and $P_X(x)$ and $P_Y(y)$ are the marginals: $P_X(x) = \Sigma_y P_{XY}(x,y)$.

**3.2.2 SHAP values.** The SHAP (SHapley Additive exPlanations) technique, which was proposed by Lundberg and Lee in 2017 [41], utilizes Shapley values derived from game theory [42] to offer explanations for models at the level of individual data points. Shapley values represent a theoretical construct derived from the field of cooperative game theory, which serves to quantify the individual contributions of several features towards the prediction of a specific occurrence.

The expression of Shapley values is as follows:

$$\varphi_i = \sum_{S \subseteq N\{i\}} \frac{|S|!(M - |S| - 1)!}{M!} (f_x(S \cup i) - f_x(S))$$

Where:

- $\phi\_i(f)$ represents the Shapley value for feature i and model f.

- N is the set of all features.

- S is a subset of features, excluding feature i.

- f(S) is the model's prediction when considering the subset of features **S**.

- f(S∪{i}) is the model's prediction when including feature **i** along with the subset of features **S**.

By conducting computations of the Shapley values pertaining to each feature, valuable insights can be obtained into the individual contributions made by features towards the predictive outcomes of the model for a given data point. This facilitates the ability to view and comprehend the decision-making process of the model at an individual level. The computation of SHAP values involves the analysis of model predictions under two scenarios: one where a certain feature is included and another where it is excluded. The present study employs a comparative analysis methodology that involves a systematic evaluation of all potential combinations of attributes pertaining to the specific issue under investigation. The computation necessitates the application of combinatorial calculus and involves the requirement to retrain the model for every combination. The process of calculating Shapley values for individual variables and subsequently averaging them across the dataset facilitates the determination of "global" values. These values provide an assessment of the overall significance or pertinence of each variable, while also enabling the evaluation of the influence of individual features on specific instances.

## 4. Our approach

This part presents a thorough examination of the many stages carried out in our research. The paper encompasses a comprehensive description of dataset preparation techniques, the methodologies applied for feature selection to identify pertinent features, the implementation settings for our proposed model (Fig 3), a comprehensive presentation of experimental results, and following in-depth discussion and analysis of the findings.

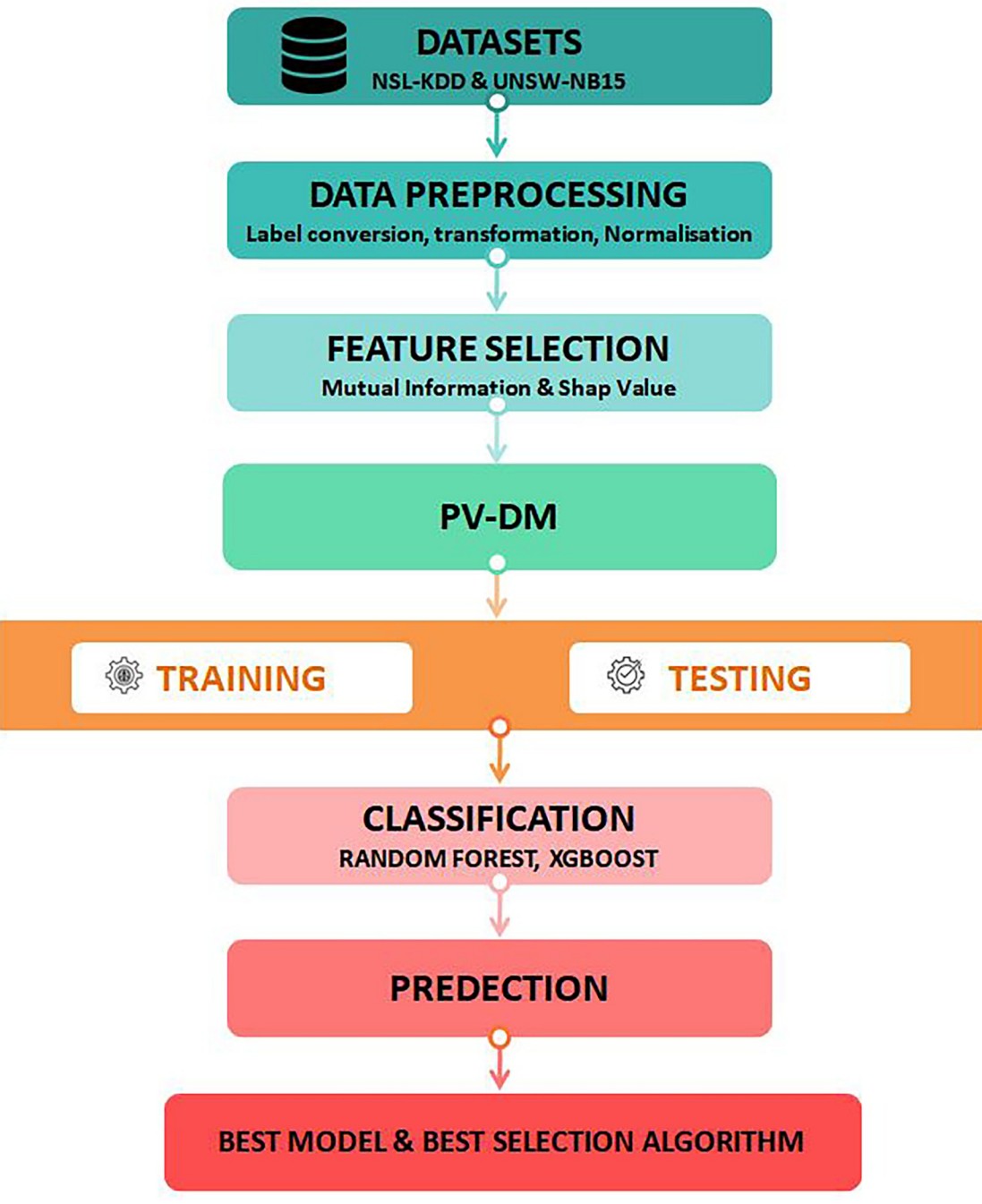

**Fig 3. Our proposed approach.**

## 4.1 Datasets

The present study employs two datasets, specifically the NSL-KDD Dataset [43] and the UNSW NB15 [44], in order to undertake experiments that aim to validate our suggested approach.

**4.1.1 NSL-KDD dataset.** The NSL-KDD dataset can be regarded as an enhanced and contemporary iteration of the KDD-Cup 99 dataset [45]. The dataset is deemed more credible for analysis purposes as it lacks redundant or duplicate records. The original NSL-KDD dataset comprised a total of 4,898,430 records, with 972,780 being classified as normal records and 3,925,650 classified as attack records. The collection comprises a total of 38 distinct categories of threats, which encompass Smurf, Neptune, devil, ipsweep, and portsweep, among others. The NSL-KDD dataset consists of a total of 43 features, which can be classified into four distinct groups: Categorical (features 2, 3, 4, and 42), Binary (features 7, 12, 14, 20, 21, and 22), Discrete (features 8, 9, 15, and features 23 to 41, and 43), and Continuous (features 1, 5, 6, 10, 11, 13, 16, 17, 18, and 19).

**4.1.2 UNSW-NB15: (University of New South Wales–NB 2015).** The creation of the UNSW-NB15 dataset [44] took place in 2015 at the Australian Centre for Cyber Security (ACCS), with the primary objective of facilitating research on intrusion detection. The data was obtained by employing an IXIA tool to collect a combination of regular and malicious actions. The dataset comprises 49 features that have been classified into five categories: Binary, Float, Integer, Nominal, and Timestamp. It spans a broad spectrum of typical and aberrant behaviors for thorough investigation.The UNSW-NB15 dataset encompasses a range of attack types, totaling nine distinct categories. These categories consist of Fuzzers, Analysis, Backdoors, Denial of Service, Exploits, Generic, Reconnaissance, Shell code, and Worms. The utilization of this dataset has been extensively embraced by scholars in the domain of intrusion detection system evaluation.One notable observation pertaining to the dataset is that the quantity of assault records surpasses the number of normal records. Moreover, there exists a notable disparity in the number of records among different types of attacks, leading to a data imbalance concern. The uneven distribution of data in the dataset can provide difficulties in efficiently training and evaluating intrusion detection systems.The graphical representation of the distribution of attack and normal records in the UNSW-NB15 and NSL-KDD datasets is depicted in Fig 4A and 4B, respectively.

Due to the small number of records for specific types of attacks in both datasets, as previously explained and illustrated in Fig 4A and 4B, we opted to focus our research on the initial four attack categories in each dataset, as indicated in Table 2.

## 4.2 Datasets preprocessing

The process of data preprocessing is a critical and fundamental step in the construction of a machine-learning model. Data preprocessing refers to the process of preparing raw, unprocessed data in order to make it suitable for analysis and model training purposes. The process encompasses a variety of methodologies, such as data cleansing, data transformation, and feature selection. The primary objective of this procedure is to ensure that the data is structured in a way that enables effective usage by machine learning algorithms.

**4.2.1 Categorization of attacks.** The attacks present in the NSLKDD dataset have been classified into four distinct categories, as depicted in Table 3.

**4.2.2 Encoding labels and numericizing features.** During this phase, we conducted label encoding for each sort of attack. In the NSL-KDD dataset, we allocated a numeric representation of 0 to the category labeled as "Normal," 1 to "Denial of Service" (DoS), 2 to "Probe," 3 to "Remote-to-Local" (R2L), and 4 to "User-to-Root" (U2R). For the UNSW-NB15 dataset, the

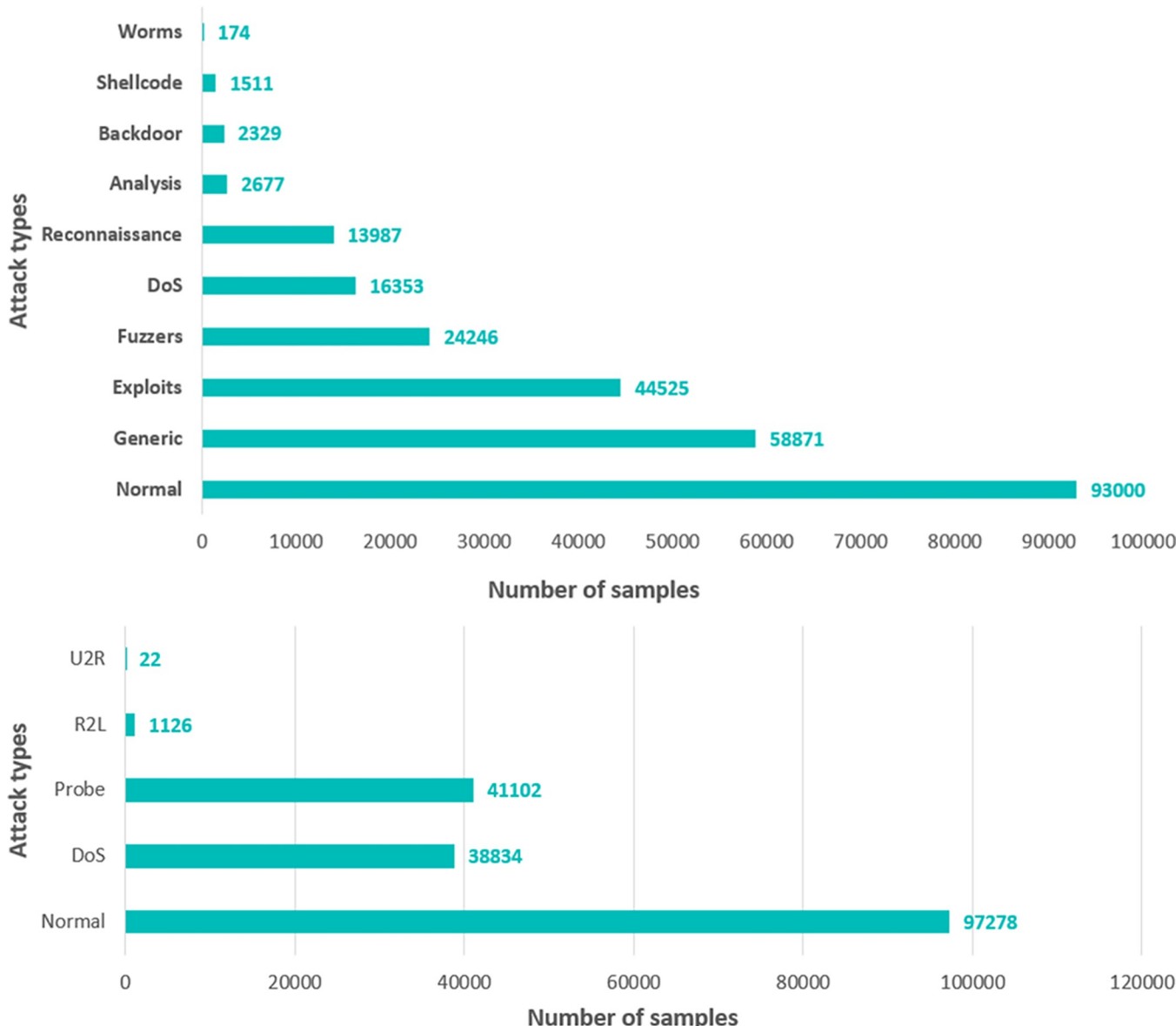

**Fig 4. a.** Attack and normal records in the UNSW-NB15. **b.** Attack and normal records in the NSL-KDD.

values assigned to the categories are as follow: Normal is represented by 0, Generic by 2, Exploits by 3, and Fuzzers by 4. Furthermore, many machine learning algorithms lack support for categorical data, necessitating its conversion into numerical form prior to utilization in the models. Hence, the non-numeric attributes ('protocol-type, service, and flag' for NSL-KDD and 'proto, service, and state' for UNSW-NB15) were converted into numerical

**Table 2. The disparate attack categories in the two datasets.**

| Dataset | Categories of attacks |
|---|---|
| NSL-KDD | Normal, DoS, Probe, R2L |
| UNSW-NB15 | Normal, Generic, Exploits, Fuzzers |

**Table 3. Attacks categories in the NSLKDD dataset.**

| Attack Category | description | Attack type |
|---|---|---|
| DoS | Denial of service is a malicious attack aimed at preventing normal users from using a service by making it inaccessible. | neptune, back, land, pod, smurf, teardrop, udpstorm, mailbomb, apache2, processtable, worm |
| Probe | Probe or surveillance is an attack that her purpose is to obtain important information about the security of networks in order to change with the security settings. | ipsweep, nmap, portsweep, satan, mscan, saint |
| R2L | This class of attacks tries to gain local unauthorized access to a remote machine by sends packets to the network. | ftp_write,guess_passwd,imap,multihop,phf,spy, warezclient,warezmaster,sendmail,named, snmpgetattack,snmpguess,xlock,xsnoop,httptunnel |
| U2R | The primary purpose of this attack is to illegally explore or steal data, install viruses, or cause the sufferer to suffer harm by introduction a normal user account. | buffer_overflow, Loadmodule, perl, rootkit, ps, sqlattack, xterm |

representations using the pandas factorize () function. This facilitated the representation of categorical variables through the use of appropriate numeric labels that are compatible with machine learning methods.

**4.2.3 Normalization.** Standardization, additionally referred to as normalization, is a statistical method employed to scale data by centring it around the mean and adjusting it to possess a standard deviation of one unit. The process of standardization is of utmost importance due to the presence of disparate characteristics within the dataset, including but not limited to time, source bytes, and destination bytes, which may exhibit significant variations in their respective maximum and minimum values. In order to accomplish this, the StandardScaler algorithm [46] is employed to compute the standardized values.

$$z = (x_i - mean(x))/stdev(x)$$

Where mean(x) represents the mean of the training sample, stdev(x) represents the standard deviation of the training sample, and xi represents the value of the specific feature.

## 4.3 Experimental results and discussion

The programming language used in this study was Python 3.7.1, and the integrated development environment employed was Google Colab. The implementation utilized several libraries, such as NumPy, Pandas, Keras, and Scikit-Learn (Sklearn). The pre-processed datasets from UNSW-NB15 were partitioned into a training set comprising 70% of the data and a testing set including the remaining 30%. The division of the NSL-KDD dataset was specifically designed to account for the varying number of attack records. In particular, 30% of the dataset was allocated for testing purposes, encompassing the DoS, Normal, and Probe categories. Conversely, due to its relatively limited quantity of 1126 records, the R2L category was assigned a smaller portion of 10% for testing.

**4.3.1 Feature selection results.** The selection of features plays a crucial role in the data preprocessing phase [47]. The proposed methodology aims to improve the accuracy and effectiveness of machine learning models by focusing specifically on relevant data and prioritizing its quality.

Figs 5 and 6 depict the visual representation of the data obtained from the UNSW-NB15 and NSL-KDD datasets, respectively, within a reduced-dimensional space. This reduction is achieved by employing Principal Component Analysis (PCA) as a data transformation

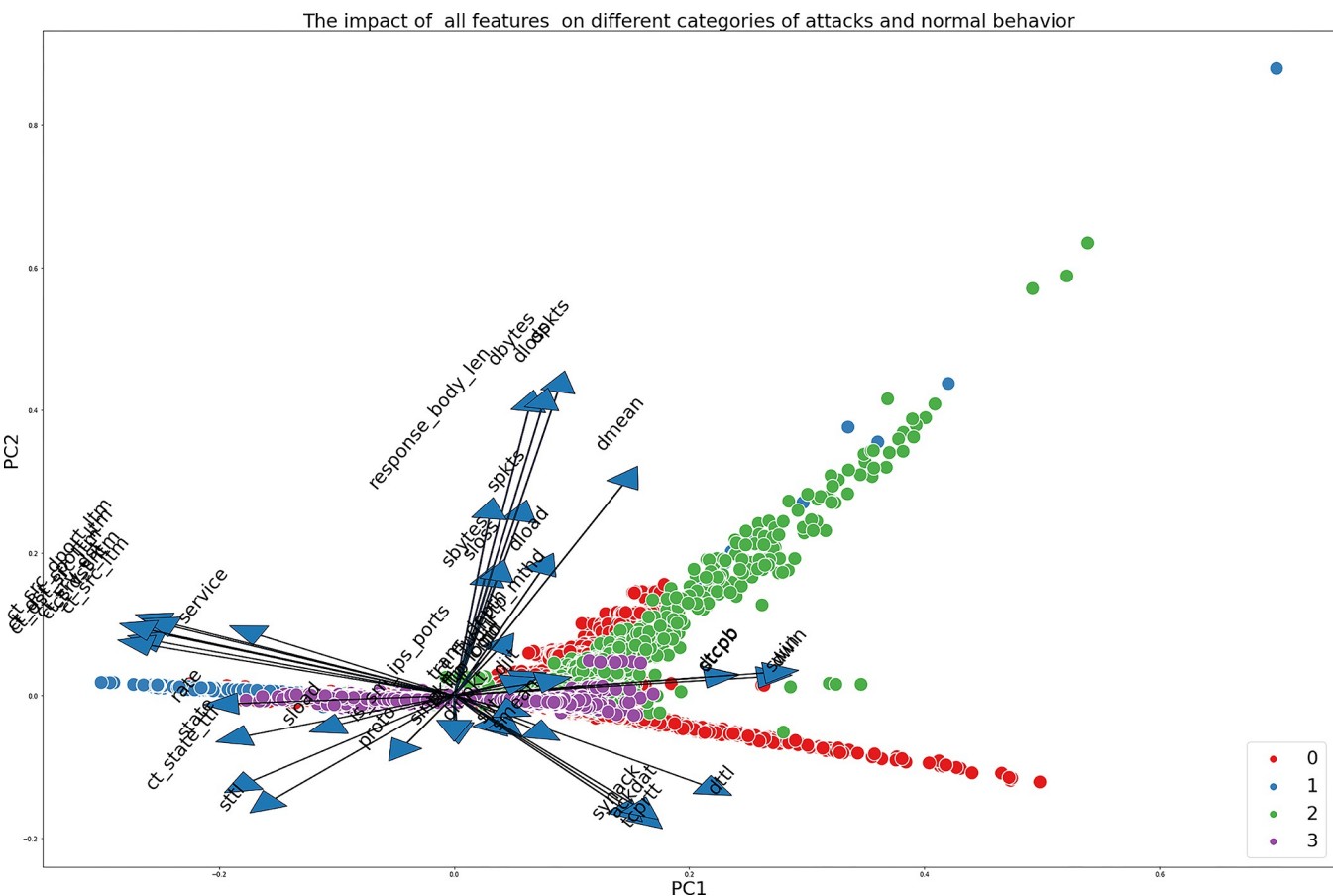

**Fig 5. The Impact of all features on different categories of attacks and normal behaviour for UNSW-NB15.**

technique. The graphs illustrate the effects of various features on different types of attacks. It is apparent that specific attributes, such as Djit, smean, and dinpkt in the UNSW-NB15 dataset, have a reduced degree of significance in comparison to other attributes. Hence, the application of attribute reduction methods becomes imperative to reduce non-informative or confusing data.

As mentioned earlier, mutual information quantifies the level of dependence between two variables. In this study, we utilize mutual information to assess the impact of each feature on the target variable in both datasets.

As depicted in Fig 7A, the features src_bytes, service, count, dst_bytes, srv_count, logged_in, dst_host_diff_srv_rate, rate, dst_host_srv_count, dst_host_same_src_port_rate, and dst_host_srv_diff_host_rate have the highest mutual information scores in the NSL-KDD dataset. On the other hand, it can be observed from Fig 7B that the attributes sbytes, sload, smean, dbytes, rate, dur, ct_dst_sport_ltm, dmean, ct_state_ttl, and service in the UNSW dataset exhibit the greatest scores for mutual information. On the other hand, the remaining features in both datasets demonstrate low scores, suggesting that they have limited observable influence on the target variable.

On the contrary, as previously stated, SHAP values offer insights into the individual contributions of each feature in predicting the target variable. To evaluate the importance of a particular feature, the model considers the results of every conceivable combination of features, as depicted in Fig 8.

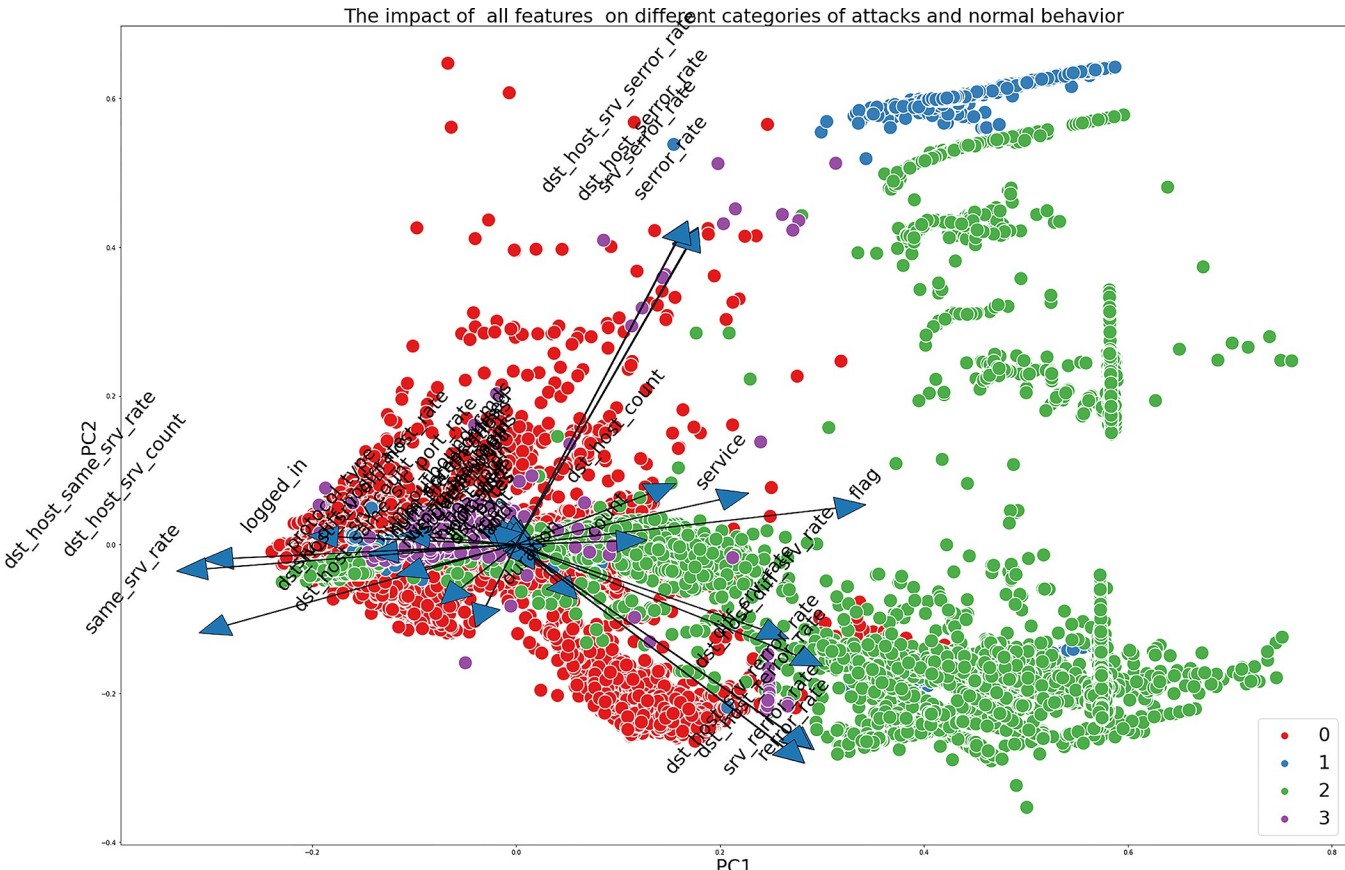

**Fig 6. The Impact of all features on different categories of attacks and normal behaviour for NSL-KDD.**

According to the visual representations in Fig 8A and 8B, it can be observed that the NSL-KDD dataset demonstrates the most prominent SHAP values for the features "src_bytes, dst_bytes, service, dst_host_diff_srv_rate, dst_host_serror_rate, rerror_rate, dst_host_rerror_-rate, dst_host_same_src_port_rate, count, and hot". In contrast, the UNSW-NB15 dataset highlights the following as its top 10 most notable features: "sttl, sbytes, ct_dst_src_ltm, proto, dur, ct_srv_dst, ct_state_ttl, response_body_len, sloss, and dbytes". The influence of these factors on the target prediction has been determined based on their respective SHAP values. Additionally, it can be noticed in Fig 7A and 7B that the Mutual Information technique used in the NSL-KDD dataset highlights several features, namely srv_count, logged_in, dst_host_srv_count, and dst_host_srv_diff_host_rate, as being among the top 10 features with significant scores. Nevertheless, the SHAP values do not assign a high ranking to these characteristics inside the top ten. Instead, features such as dst_host_serror_rate, rerror_rate, dst_host_serror_rate, and hot are given higher rankings. The observed disagreement suggests that the assessment of feature relevance can differ across Mutual Information and SHAP values, hence emphasizing the distinct perspectives and criteria employed by these methodologies.

Following the utilization of the Mutual Information and SHAP values algorithms to identify the most significant features from both datasets, a total of four reduced datasets were derived. The datasets consist solely of the top 10 features that were determined by the algorithms in consideration. The summary of these four reduced datasets is provided in Table 4.

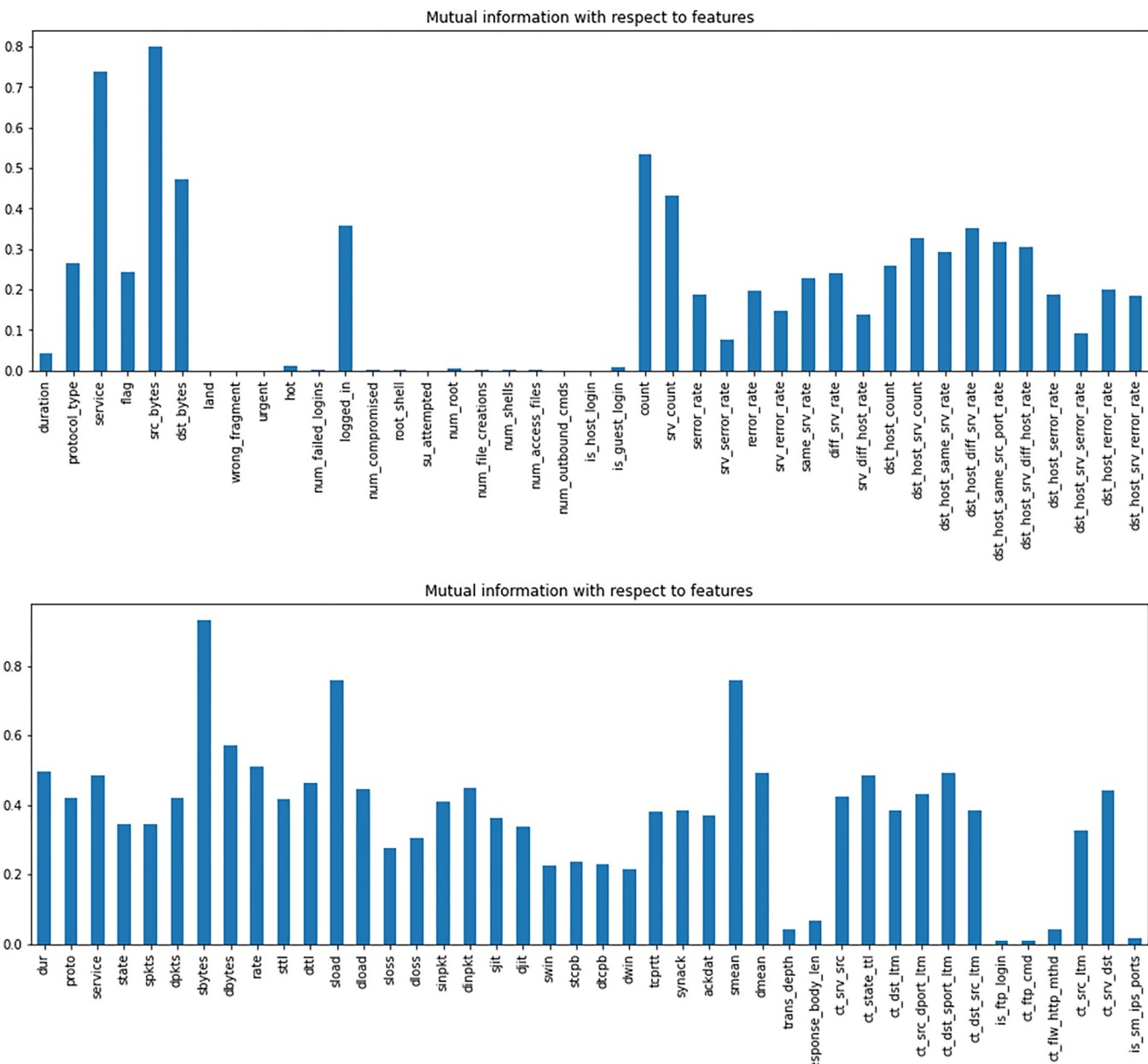

**Fig 7. a.** NSL-KDD feature Mutual Information scores. **b.** UNSW-NB15 features Mutual Information scores.

Figs 9A, 9B, 10A and 10B depict the influence of the selected characteristics utilizing the Mutual Information (MI) and SHAP values on the two datasets.

Both datasets demonstrate that certain features have a substantial influence on particular sorts of attacks. In the context of the UNSW-NB15 dataset specifically focusing on attack 2 (Exploits), it is evident that certain features, namely response_body_len, dpkts, sbytes, and spkts, exhibit a notable impact on the detection of this attack. This observation underscores the significance of these attributes in effectively identifying and comprehending this specific form of attack.

**4.3.2 Model of shallow learning results.** Following the completion of the feature reduction phase, we employ the PV-DM (Paragraph Vector-Distributed Memory) model to

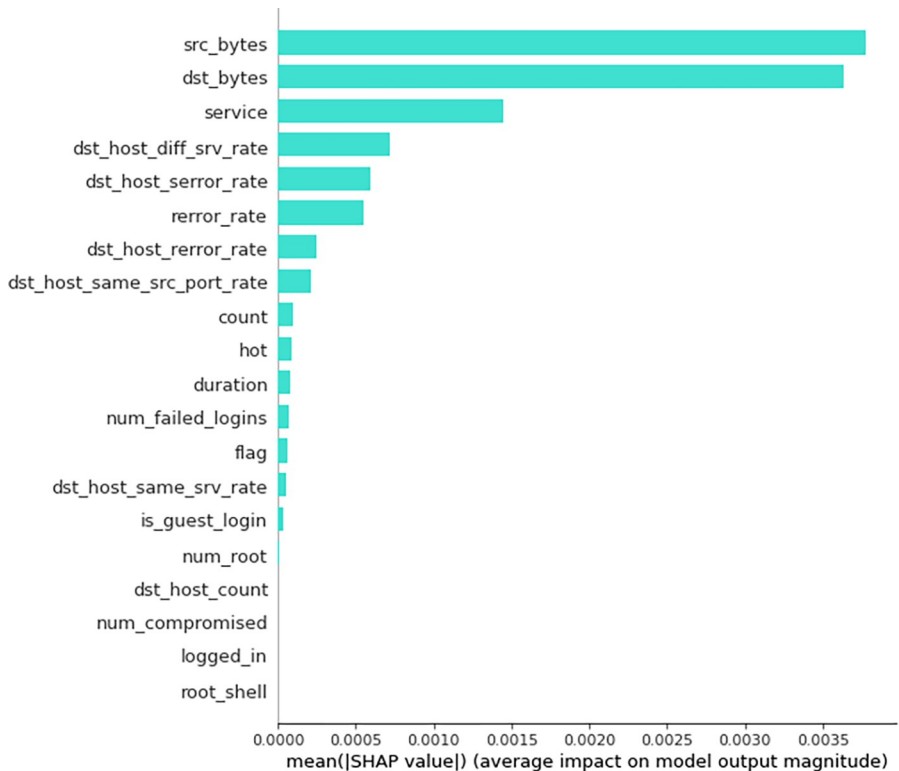

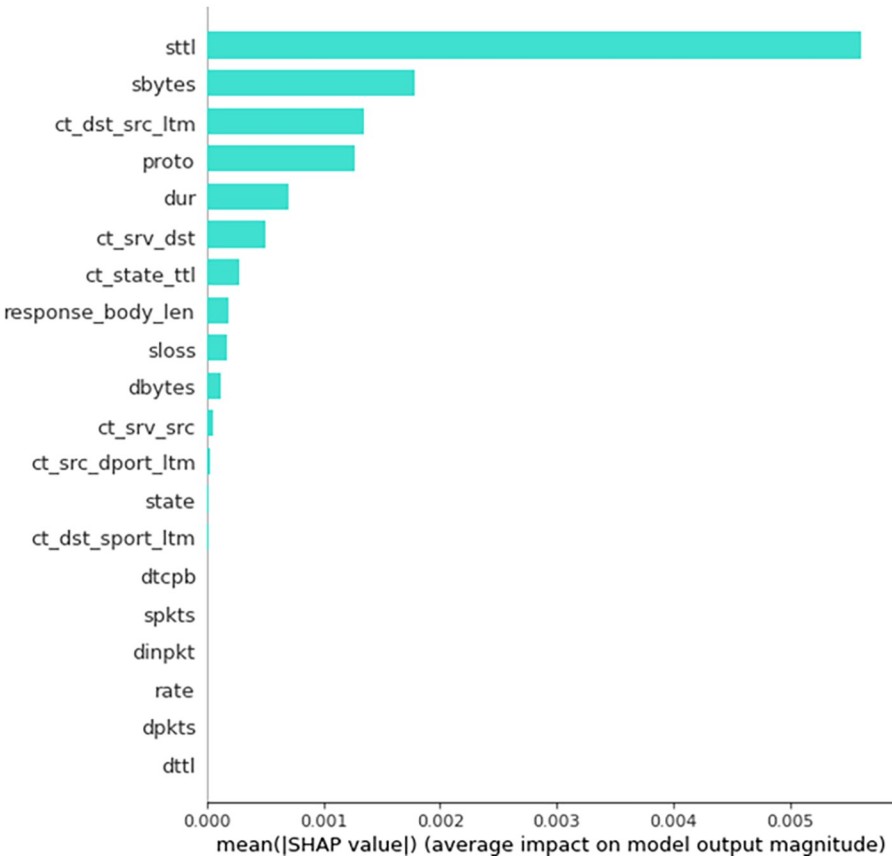

**Fig 8. a.** Significance of features within the NSL KDD dataset using SHAP. **b.** Significance of features within the UNSW15dataset using SHAP.

construct embedding vectors. The aforementioned vectors will serve as inputs for the machine-learning models. A total of six sets of embedding vectors are produced, comprising four sets derived from the reduced datasets specified in Table 7, and two sets obtained from the original datasets (NSL-KDD and UNSW-NB15), which encompass all the features. The PV-DM model is created with the parameters stated in Table 5.

The aim of our study is to enhance our approach by determining the most precise feature selection method and classifier, utilizing performance metrics including accuracy, precision, recall, F1-score, and AUC (as depicted in Fig 11). In order to accomplish this, we partitioned the PV-DM embedding vectors of each set into separate sets for training and testing purposes, as shown in Table 6. The Random Forest and XGBoost classifiers were employed to classify the data, as they have demonstrated their efficacy in the categorization of attacks in several studies, such as those referenced in [48, 49].The results of these classifications are presented in Figs 11 and 12.

The performance metrics, namely accuracy, recall, precision, F1-score, and AUC, for each model are depicted in Figs 11 and 12. Additionally, Fig 13 depicts the Receiver Operating Characteristic (ROC) plot for each classifier.

As an initial observation, it is evident that the inclusion of pertinent data enhances the accuracy of attack categorization, as demonstrated by the obtained findings. Specifically, employing Mutual Information (MI) or Shap value techniques lead to higher scores in assessment metrics, in contrast to classifying the data based on all available features. Moreover, based on the findings depicted in Figs 11 and 12, it is evident that the PV-DM with the XGBOOST classifier consistently demonstrated better performance in terms of accuracy, recall, and F1 score for both datasets when compared to the RANDOM-FOREST classifier. The assessment of these metrics holds significant significance in evaluating the efficacy of classification results. Additionally, it is important to highlight that the PV-DM model, when coupled with the random forest algorithm, demonstrated notable precision scores, as indicated by both the mutual information (MI) and SHAP values. In contrast, it is evident that when comparing the efficacy of MI and SHAP values as feature selection techniques, the consistent utilization of SHAP values leads to superior performance outcomes. While Mutual Information (MI) may not exhibit the same level of performance as SHAP values, it is apparent that the inclusion of Mutual Information in PV-DM enhances performance across several metrics and classifiers in comparison to utilizing PV-DM with all features.

In order to emphasize the classification performance of our proposed intrusion detection system, we provide a visual representation of the Receiver Operating Characteristic (ROC) plot for each classifier, as shown in Fig 13. Based on the presented plots, it can be observed that our suggested model exhibits a commendable ability to effectively reduce the occurrence of attacks that are erroneously labeled as normal behavior, while successfully identifying and

**Table 4. Features of reduced datasets.**

| Reduced Dataset | Features Included |
|---|---|
| Red_NSL_MI | 5, 3, 23, 6, 24, 12, 35, 33, 36, 37 |
| Red_NSL_ShapV | 5, 6, 3, 35, 38, 27, 40, 36, 23, 10 |
| Red_UNSW_MI | 4, 9, 24, 5, 6, 1, 32, 25, 29, 41 |
| Red_UNSW_ShapV | 7, 4, 33, 40, 1, 38, 29, 27, 11, 5 |

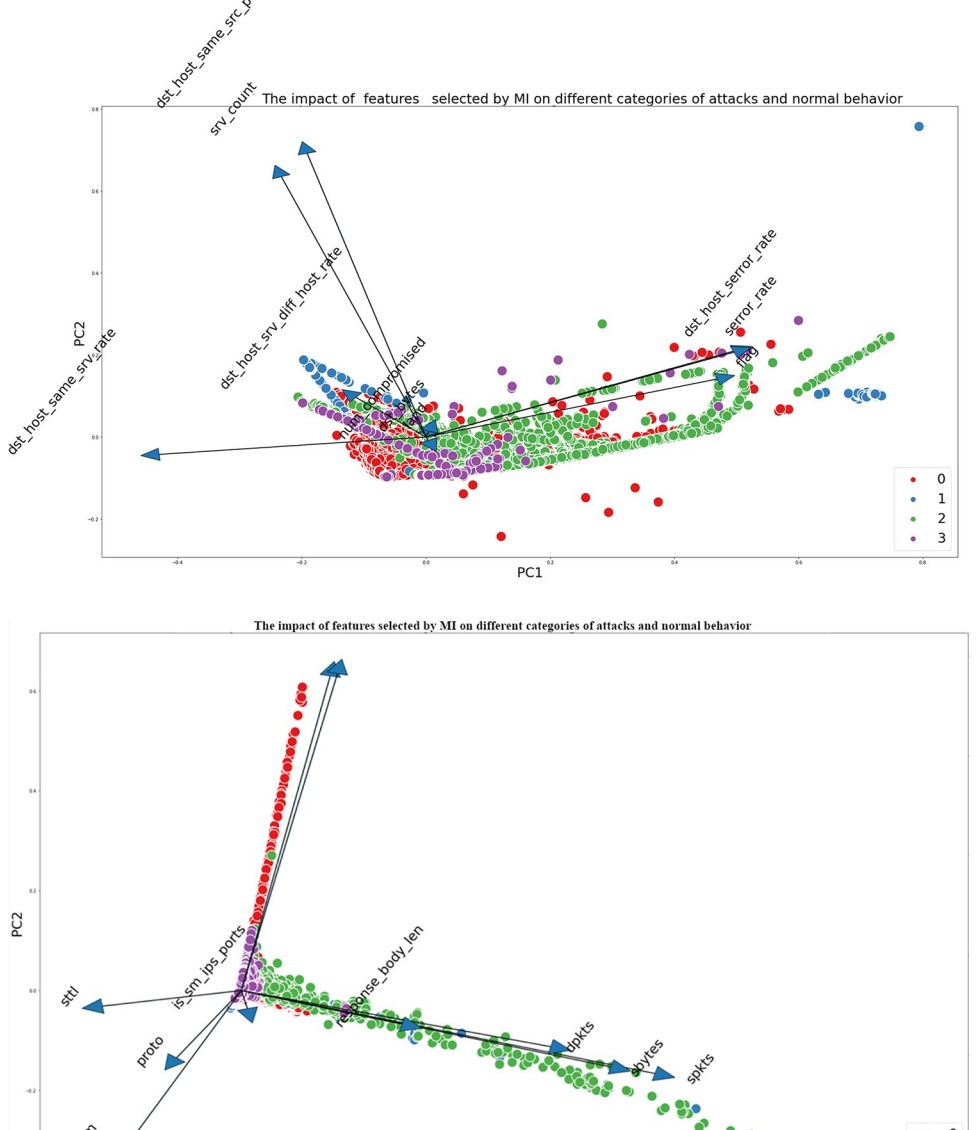

**Fig 9. a.** The Impact of the selected features by MI on different categories of attacks and normal behavior for NSL-KDD dataset. **b.** The Impact of the selected features by MI on different categories of attacks and normal behavior for UNSW-NB15.

capturing unusual attacks. This characteristic holds significant value in practical scenarios, particularly for IT professionals.

**4.3.3 Comparison of our approach versus LSTM.** In this section, a comparative analysis will be conducted between the proposed model and the LSTM algorithm, which is widely acknowledged as a highly efficient and extensively utilized method for intrusion detection. The purpose of this comparative analysis is to assess the efficacy of our model, with particular attention to the reduction of false negative occurrences. LSTM networks were implemented

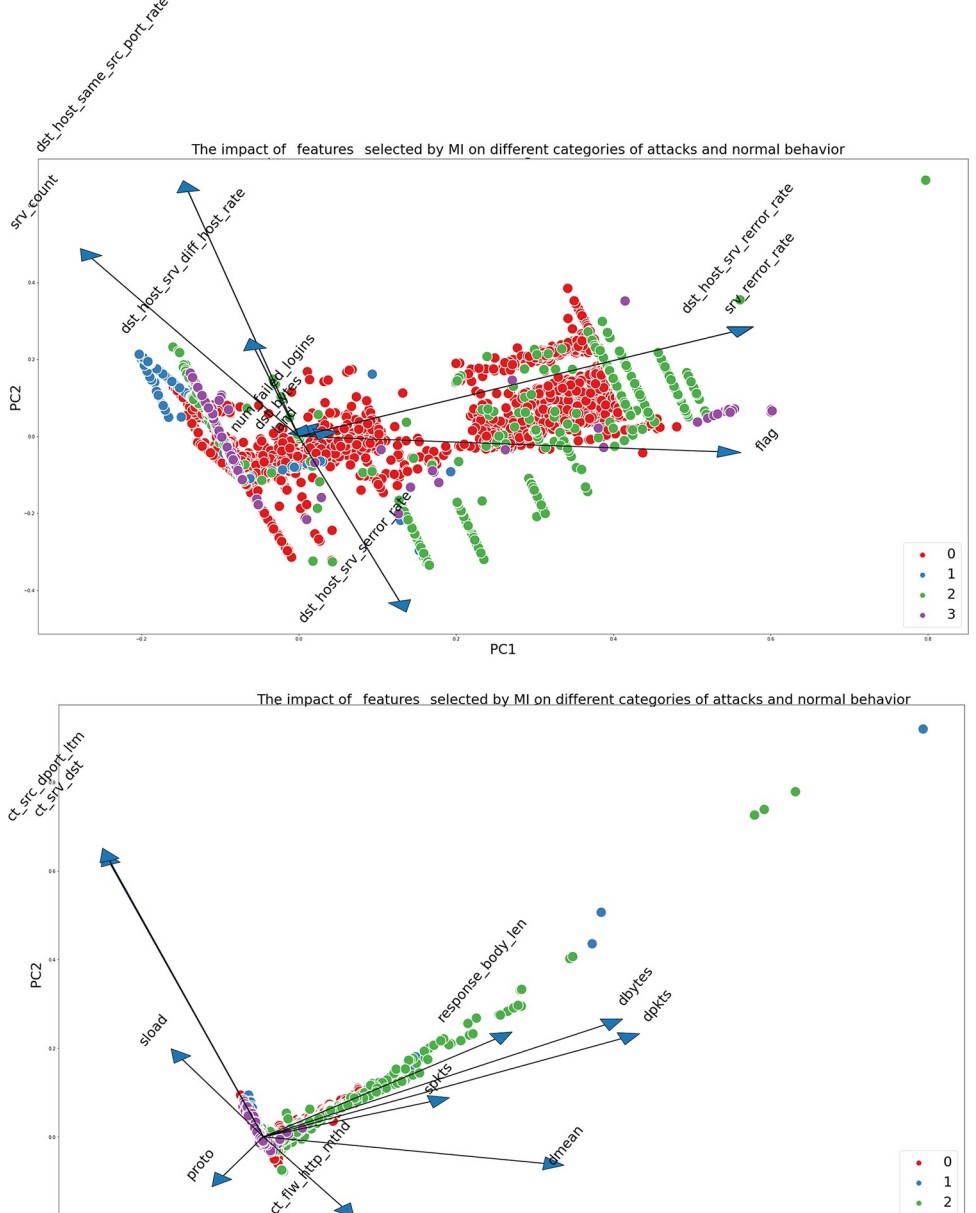

**Fig 10. a.** The Impact of the selected features by Shap Value on different categories of attacks and normal behavior for NSL-KDD. **b.** The Impact of the selected features by Shap Value on different categories of attacks and normal behavior for UNSW-NB15.

for the purpose of pattern identification, utilizing the reduced sets as previously described in Table 5. The architectural design of these networks consisted of a single hidden layer. The parameter values for the proposed LSTM model are presented in Table 7.

LSTM demonstrated the highest level of accuracy when employing the 10 fundamental attributes identified through Mutual Information (MI) for both the NSL-KDD and UNSW-NB15 datasets (Figs 14 and 15). However, our model consistently achieved the highest levels of precision, recall, and F1 score. Additionally, when utilizing SHAP values as a dimensionality reduction technique in conjunction with LSTM, our proposed methodology

**Table 7. LSTM model parameters.**

| Parameters | LSTM values |
|---|---|
| Activation function | Softmax |
| Loss function | Sparse categorical cross-entropy |
| Optimiser | Nadam |
| Learning rate | 0.002 |
| Epsilon | 1e-08 |
| Schedule decay | 0.004 |
| Epoch number | 20 |
| Dropout | 0.3 |

exhibited the highest levels of accuracy, precision, recall, and F1-score for the reduced UNSW-NB15 dataset. On the contrary, it was noted that the performance of the Long Short-Term Memory (LSTM) model on the NSL-KDD dataset was comparable, with the exception of the accuracy metric where LSTM outperformed our methods significantly.

The findings suggest that the performance of LSTM is adversely affected when extraneous attributes are eliminated, and only pertinent features are chosen. This observation implies that Long Short-Term Memory (LSTM) models encounter difficulties in effectively distinguishing between various types of attacks. One plausible reason for this phenomenon is that the Long Short-Term Memory (LSTM) model, which belongs to the category of recurrent neural networks, necessitates a substantial volume of data to acquire a comprehensive understanding of potential risks. Moreover, the Long Short-Term Memory (LSTM) model regards each sequence as a distinct entity, which may result in a decline in performance when employing feature selection techniques.

In contrast, our proposed methodology, which utilizes the PV-DM algorithm, incorporates both the global and local context of each sequence. Even in the presence of imbalanced datasets, our approach demonstrates enhanced efficacy in detecting attacks. This underscores the efficacy of our methodology in effectively detecting and analyzing attacks by considering contextual factors.

The performance of our suggested architecture, specifically in terms of recall value, demonstrates more effectiveness in comparison to LSTM when feature selection methods are employed. The aforementioned stage is of utmost importance in the pursuit of enhancing the precision of detecting different forms of attacks, while concurrently diminishing the duration required for training and testing and augmenting the overall rate of intrusion detection [50].

**Table 5. Table of parameters of the PVDM.**

| Parameters of PVDM Model | values |
|---|---|
| Number of features | All,10 |
| min_count | 5 |
| window | 8 |
| vector_size | 100 |
| sample | 1e-4 |
| negative | 5 |
| workers | 4 |
| alpha | 0.025 |
| min_alpha | 0.025 |
| epochs | 100 |

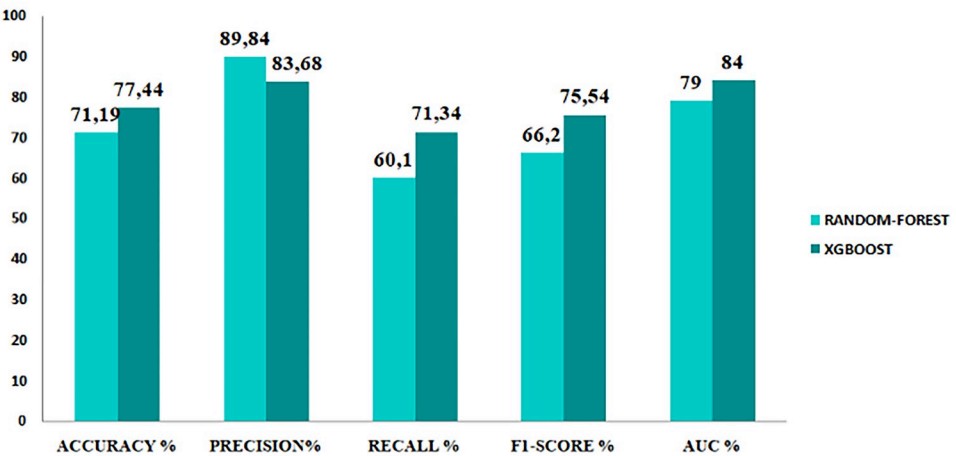

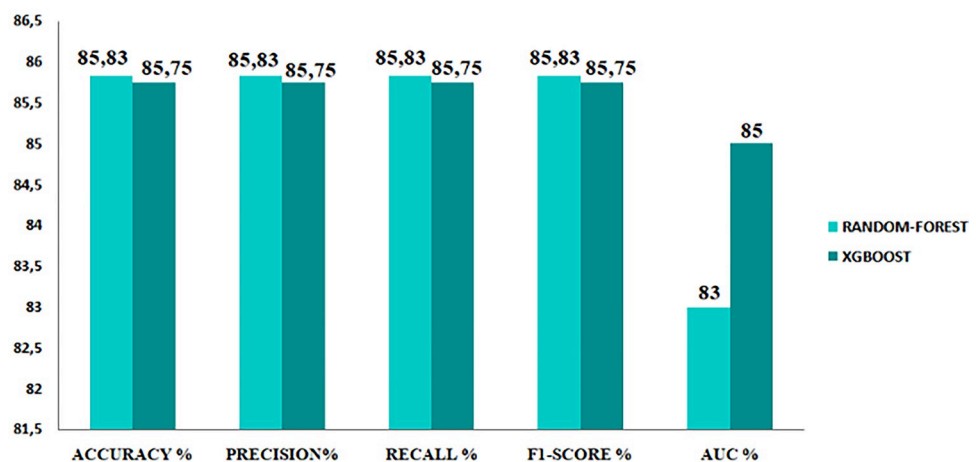

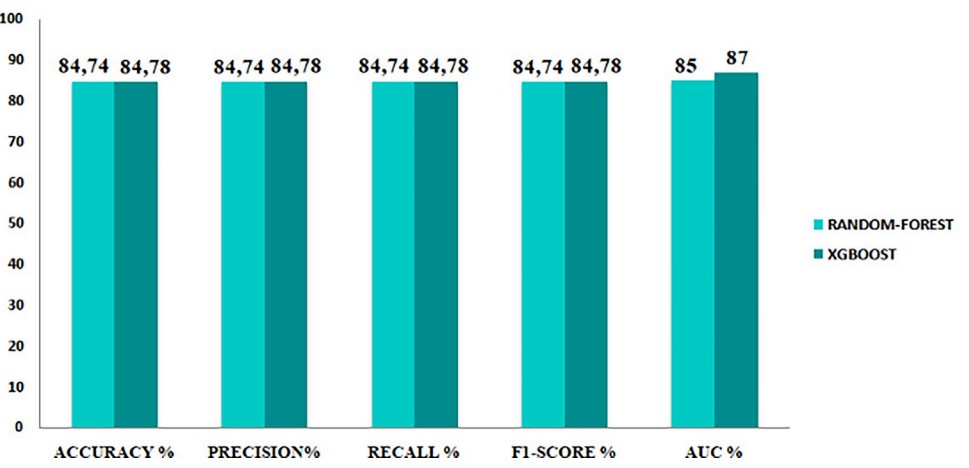

**Fig 11. Performances of classification on UNSW15 sets.**

**Table 6. The size of the training set and testing set for each set.**

| Embedding vectors of | Train size | Test size |
|---|---|---|
| Red_NSL_MI | 125061 | 53279 |
| Red_NSL_ShapV | | |
| NSL-KDD | | |
| Red_UNSW_MI | 176514 | 44128 |
| Red_UNSW_ShapV | | |
| UNSW-NB15 | | |

**4.3.4 Assessing the performance of the model via additional feature reduction.** To enhance the assessment of our model's performance, we have opted to reduce the number of features to 6 and 4, respectively, based on their corresponding scores derived from the SHAP Value methodology. The decrease in the number of features will enable us to evaluate the ability of our model to sustain its performance in the face of these altered circumstances. Based on the results shown in Table 8 of the NSL KDD dataset analysis, it is evident that our model's performance demonstrates a significant improvement in accuracy and precision when the number of features is lowered to 6 and 4, as compared to the results obtained with 10 features.

Based on the results presented in Table 9, it is evident that there is a marginal increase in accuracy for UNSW 15, with a recorded value of 82.86%. The precision value is seen to be 84.07%, while the recall value is 77.70% when utilizing a feature set consisting of 6 features. Conversely, while employing a feature set of 4 features, the model exhibits the highest accuracy rate of 83.27%, a precision rate of 83.55%, and constantly maintains a recall rate of 77.70%. The results suggest that both the 4-feature and 6-feature setups exhibit similar performance to the 10-feature configuration. This underscores that reducing the number of features from 10 to either 6 or 4 does not have a substantial impact on the model's performance. Consequently, this offers a simplified model that utilizes a smaller set of features while still achieving satisfactory classification performance. Furthermore, while doing a comparative analysis of the performance of our approach on the NSL-KDD and UNSW-NB15 datasets, we consistently observed superior performance measures for the NSL-KDD dataset. The dissimilarity between the UNSW-NB15 dataset and the NSL-KDD dataset can be ascribed to the greater complexity of the former dataset across multiple dimensions. The inclusion of a vast array of current attack scenarios and typical network traffic features in the UNSW-NB15 dataset renders it more complex to appropriately identify. However, it should be noted that the attacks observed in the NSL-KDD dataset, along with their associated usual behaviors, may not accurately reflect the present-day landscape of threats. In addition, the complexity of the UNSW-NB15 dataset is attributed to the numerous similarities seen between normal and attack occurrences across various variables [51]. The presence of this commonality poses a challenge in discerning between benign and harmful actions, resulting in a decline in classification accuracy.

Our model exhibits consistent performance even when the number of features is lowered to 6 and 4, as demonstrated here. Nevertheless, it is crucial to recognize that there exist persistent obstacles that require attention in order to successfully execute the proposed framework in practical situations. The issues essentially pertain to two fundamental aspects: the real-time execution of the system and the comprehensibility of the vectors produced by PV-DM.

**4.3.5 Comparative analysis: Evaluating our approach in comparison to previous studies.** To augment the analysis of the experimental findings, a comparative examination was carried out between the results obtained from our suggested approach and the findings of prior investigations, as delineated in Tables 10 and 11. It is crucial to acknowledge that this comparison acts as an illustrative tool due to the potential variations in the execution

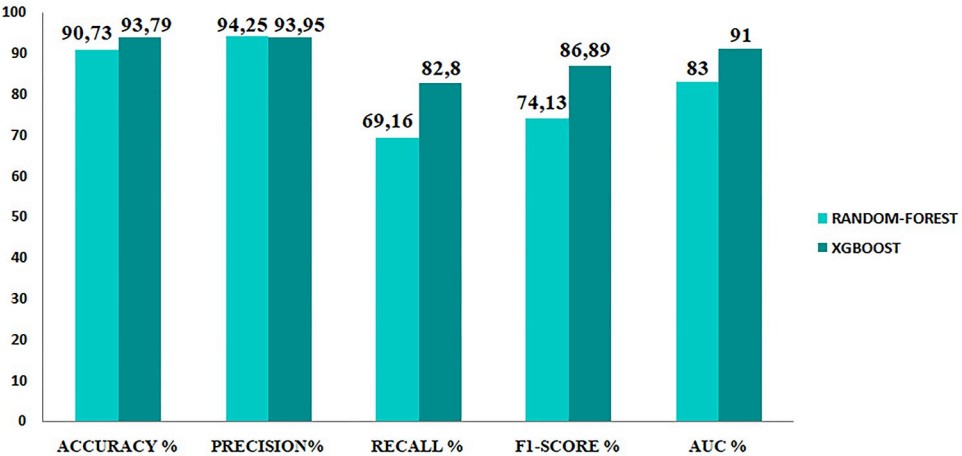

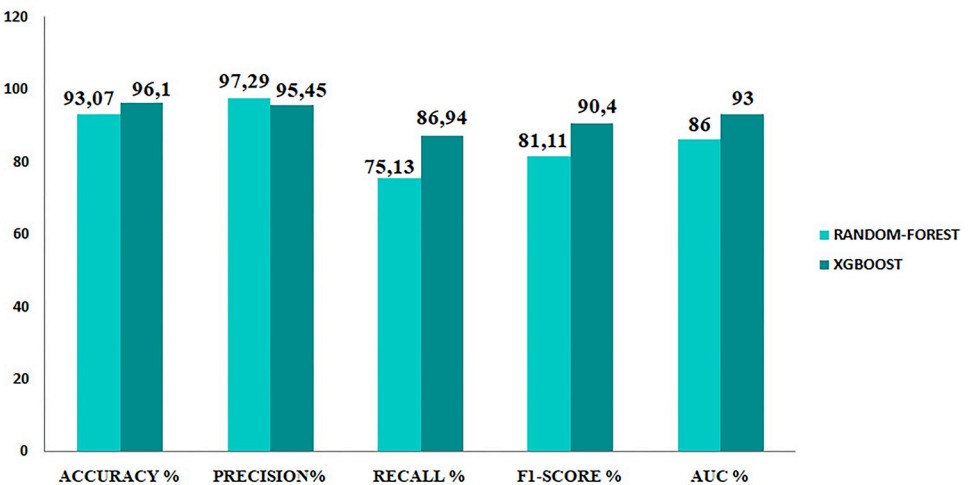

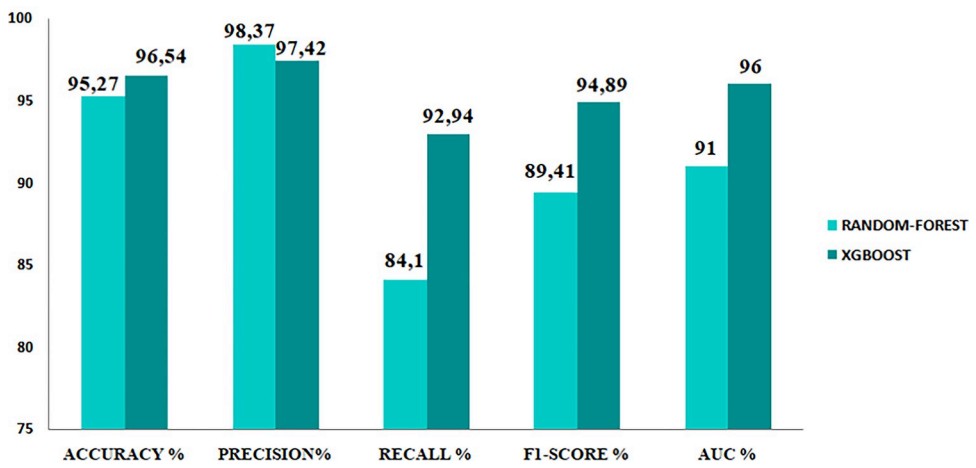

**Fig 12. Performances of classification on NSL-KDD sets.**

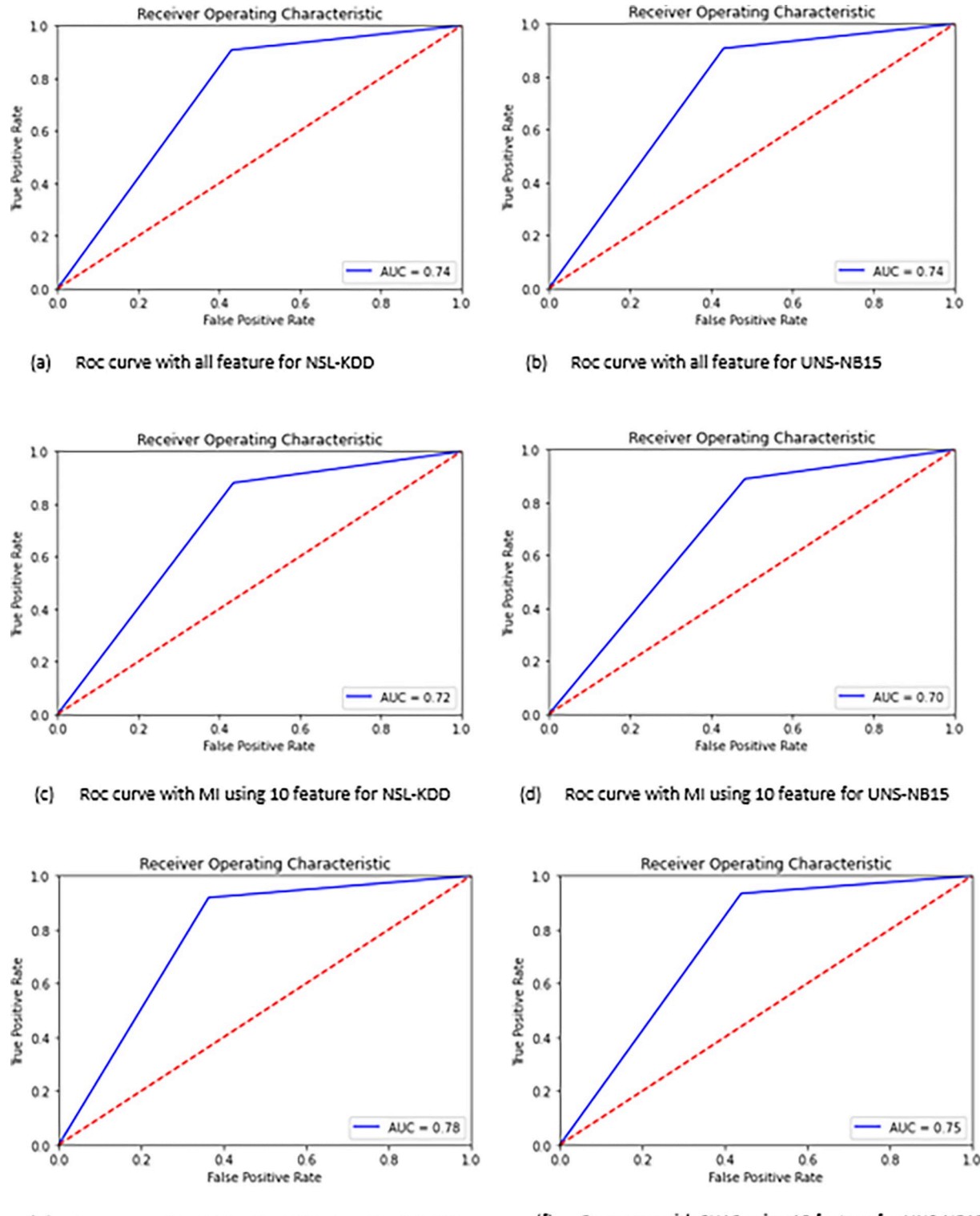

**Fig 13. The ROC curve of the XGBOOST classifier.**

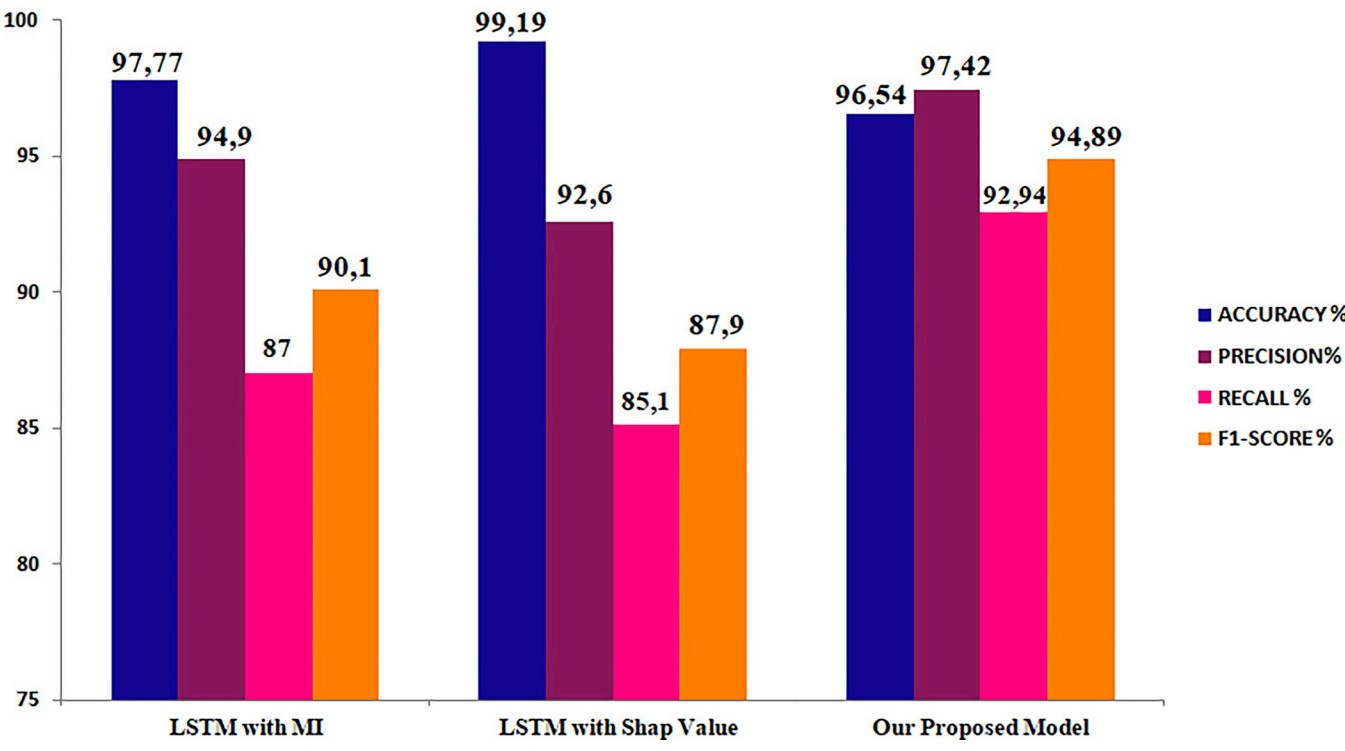

**Fig 14. Classification performance of LSTM on NSL-KDD reduced sets.**

environment, data pre-treatment procedures, and interpretation processes among Intrusion Detection Systems (IDSs).

Nevertheless, the findings of the comparison demonstrate that our model exhibits superior performance in relation to the models considered in the comparison, hence emphasizing the efficacy of our method in addressing this specific problem. This implies that our chosen methodology is more appropriate for tackling the difficulties associated with this topic.

## 5. Conclusion and future works

This paper presents a novel intrusion detection system that utilizes a shallow learning model called the Paragraph Distributed Memory (PV-DM). The method we propose incorporates the utilization of SHAP values for the purpose of feature selection, while employing the XGBoost Classifier for the execution of classification tasks. In this study, we present a comparison analysis between our technique and the Long-Short Term Memory (LSTM) methodology. Our findings indicate that our approach exhibits greater performance in attack detection, even when considering the imbalanced nature of the NSL-KDD and UNSW-NB15 datasets. Notably, our strategy does this by utilizing only relevant features. This suggests that our strategic approach demonstrates improved performance and an increased capacity to identify and respond to security breaches. Significantly, despite the reduction of features to six, resulting in an accuracy of 82.86%, precision of 84.07%, recall of 77.70%, and an F1-score of 80.20% for the UNSW-NB15 dataset, and the reduction of features to four, giving an accuracy of 98.92%, precision of 98.92%, recall of 95.44%, and an F1-score of 96.77% for the NSL-KDD dataset, our

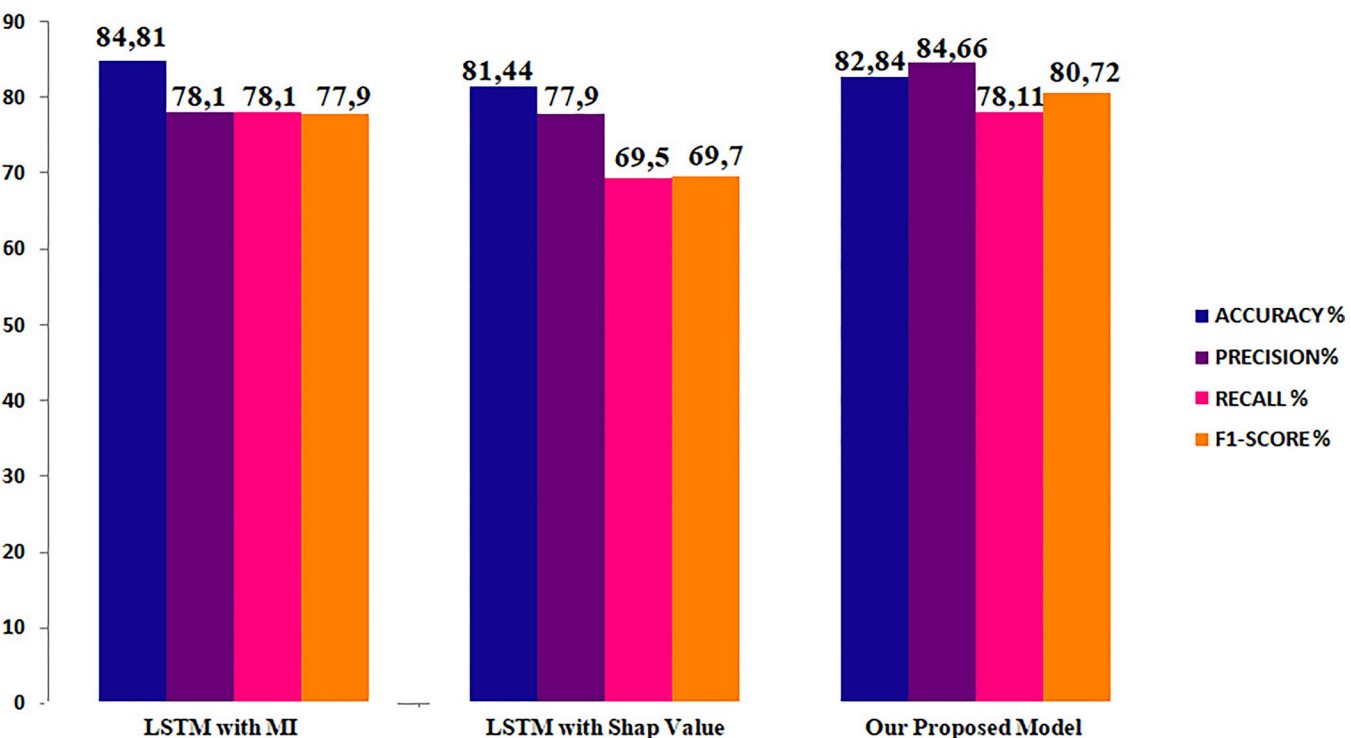

**Fig 15. Classification performance of LSTM on UNSW-NB15 reduced sets.**

**Table 8. Classification performance of our method with 6 and 4 attributes on NSL-KDD.**

| FEATURES | ACCURACY % | PRECISION% | RECALL % | F1-SCORE % |
|---|---|---|---|---|
| 06 | 98.76 | 98.86 | 95.55 | 97.10 |
| 04 | 98,92 | 98,92 | 95,44 | 96,77 |

**Table 9. Classification performance of our method with 6 and 4 attributes on UNSW-NB15.**

| FEATURES | ACCURACY % | PRECISION% | RECALL % | F1-SCORE % |
|---|---|---|---|---|
| 06 | 82.86 | 84.07 | 77.70 | 80.20 |
| 04 | 83.27 | 83.55 | 77.70 | 79.81 |

**Table 10. Comparison between different models and our approach on NSLKDD dataset.**

| Model | Feature selection method | Accuracy% | Precision% | Recall% | F1-score% |
|---|---|---|---|---|---|
| LCNN-GRNN [52] | Enhanced Shuffled Frog Leaping (ESFL) | 92.56 | - | - | - |
| DNN [53] | 04 features | 78,1 | - | - | - |
| Decision Tree with 05 features [54] | CPIO (Continuous Pigeon Inspired Optimizer) (05 features) | 88.3 | - | - | 88.2 |
| two-stageclassification algorithm (rotation forest and bagging) [55] | Hybrid (GA+PSO+ACO) (37 features) | 85,79 | 88,0 | 86,8 | - |
| Proposed model | SHAP values (04 features) | 98,92 | 98,92 | 95,44 | 96,77 |

Table 11. Comparison between different models and our approach on UNSW15 dataset.

| Model | Feature selection method | Accuracy% | Precision% | Recall% | F1-score% |
|---|---|---|---|---|---|
| IGAN-IDS [56] | Feed-forward Neural Network | 82.53 | - | - | - |
| DNN [53] | - | 64,5 | 61,4 | 64,5 | 58,6 |
| Proposed model | SHAP values(06 features) | 82.86 | 84.07 | 77.70 | 80.20 |

proposed model consistently exhibits strong and reliable performance. In future research endeavors, there will be a significant focus on investigating the integration of novel architectural designs for deep learning, such as attention mechanisms, with advanced data augmentation techniques.

## Author Contributions

**Conceptualization:** Chadia E. L. Asry, Ibtissam Benchaji, Samira Douzi, Bouabid E. L. Ouahidi.

**Data curation:** Chadia E. L. Asry.

**Formal analysis:** Chadia E. L. Asry, Ibtissam Benchaji, Samira Douzi.

**Funding acquisition:** Chadia E. L. Asry.

**Investigation:** Chadia E. L. Asry, Ibtissam Benchaji, Samira Douzi.

**Methodology:** Chadia E. L. Asry, Ibtissam Benchaji, Samira Douzi, Bouabid E. L. Ouahidi.

**Project administration:** Chadia E. L. Asry, Samira Douzi, Bouabid E. L. Ouahidi.

**Software:** Chadia E. L. Asry, Ibtissam Benchaji.

**Supervision:** Ibtissam Benchaji, Samira Douzi, Bouabid E. L. Ouahidi.

**Validation:** Chadia E. L. Asry, Ibtissam Benchaji, Samira Douzi, Bouabid E. L. Ouahidi.

**Visualization:** Chadia E. L. Asry, Samira Douzi.

**Writing – original draft:** Chadia E. L. Asry, Ibtissam Benchaji, Samira Douzi.

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
