## [Decision Letter · Decision Letter 0]

24 Oct 2023

PONE-D-23-32323Enhancing Intrusion Detection Using Shallow Learning and Feature ExtractionPLOS ONE

Dear Dr. Douzi,

Thank you for submitting your manuscript to PLOS ONE. After careful consideration, we feel that it has merit but does not fully meet PLOS ONE’s publication criteria as it currently stands. Therefore, we invite you to submit a revised version of the manuscript that addresses the points raised during the review process.

We look forward to receiving your revised manuscript.

Kind regards,

Mujeeb Ur Rehman, Ph.D.

Academic Editor

PLOS ONE

Journal Requirements:

3. Please ensure that you refer to Figure 6 in your text as, if accepted, production will need this reference to link the reader to the figure.

4. We note you have included a table to which you do not refer in the text of your manuscript. Please ensure that you refer to Table 8 in your text; if accepted, production will need this reference to link the reader to the Table.

Reviewers' comments:

Reviewer's Responses to Questions

**Comments to the Author**

1. Is the manuscript technically sound, and do the data support the conclusions?

Reviewer #1: Yes

Reviewer #2: Partly

2. Has the statistical analysis been performed appropriately and rigorously? 

Reviewer #1: Yes

Reviewer #2: Yes

3. Have the authors made all data underlying the findings in their manuscript fully available?

Reviewer #1: Yes

Reviewer #2: No

4. Is the manuscript presented in an intelligible fashion and written in standard English?

Reviewer #1: No

Reviewer #2: Yes

5. Review Comments to the Author

Reviewer #1: This manuscript have presented a detection model that utilizes a feature selection model, a shallow learning algorithm, and machine learning classifiers. Adequate revisions to the following points should be undertaken to justify the recommendation for publication.

The abstract section is fragile. Please re-write an abstract section, explain an obtained result and contribution, improve a proposed method, etc. Please delete unnecessary information.

This paper has more than spelling and grammatical errors. Please fix all of them.

The authors should clearly state the limitations of the proposed method in other real applications.

I suggest the authors add a table at the end of the literature review and compare the reviewed papers to clarify the research gap better.

The capitalization format of the title needs to be standardized. You can either capitalize the first letter of every word or use all lowercase letters.

Please re-write your contribution to this paper in the Introduction section.

Please draw a flowchart of the proposed method.

How did the authors set parameters for their proposed algorithm? Please make sensitivities of these parameters to the performance of their proposed algorithm!

Please make the Introduction and related work sections more productive using the following articles. Reading and using these articles and also cited in this article: A Multi-Objective Mutation-based Dynamic Harris Hawks Optimization for Botnet Detection in IoT, MOAEOSCA: an enhanced multi-objective hybrid artificial ecosystem-based optimization with sine cosine algorithm for feature selection in botnet detection in IoT, A Feature Selection based on the Farmland Fertility Algorithm for Improved Intrusion Detection Systems

Expand the critical results in the conclusion. Focus on the main developments in the finale. Also, write the main contributions in the conclusion.

Numerical results are good enough, but more explanations are required to analyze each figure presented.

The simulation section is not detailed enough. Authors are suggested to provide more information about their employed data and simulation process.

Please change the title of the end section (Conclusions) to (Conclusions and Future Works), and write some future works.

All figures have low quality, and please improve all of them.

Good luck

Reviewer #2: 1. Why is shallow learning combined with feature extraction used to solve detection problems, rather than end-to-end deep learning? The author should explain this issue in depth from both theoretical and practical perspectives.

2. The specific structure of machine learning models should be introduced in detail. The reasons for adopting specific model structures and feature combinations should be explained.

3. Many figures are blurry and should be re displayed.

4. Some deep learning methods should be considered for comparison.

5. Since deep learning methods have achieved remarkable results in the field of feature extraction and learning. The following related work of deep learning techniques must be cited and discussed, including “Spectrum interference-based two-level data augmentation method in deep learning for automatic modulation classification,” Neural Computing & Applications, vol. 33, pp. 7723-7745, 2020. “MR-DCAE: Manifold regularization-based deep convolutional autoencoder for unauthorized broadcasting identification,” International Journal of Intelligent Systems, vol. 36, no. 12, pp. 7204-7238, 2021. “Fine-grained modulation classification using multi-scale radio transformer with dual-channel representation,” IEEE Communications Letters, vol. 26, no. 6, pp. 1298-1302, June. 2022. Application of wavelet-packet transform driven deep learning method in PM2. 5 concentration prediction: A case study of Qingdao, China. Sustainable Cities and Society, 2023, 92: 104486. “DL-PR: Generalized automatic modulation classification method based on deep learning with priori regularization,” Engineering Applications of Artificial Intelligence, 2023, 122: 106082.

6. PLOS authors have the option to publish the peer review history of their article (what does this mean?). If published, this will include your full peer review and any attached files.

Reviewer #1: No

Reviewer #2: No

<quillbot-extension-portal></quillbot-extension-portal>

---

## [Author Response · Author response to Decision Letter 0]

24 Nov 2023

Reviewer 1

CONSIDERATIONS

1) The abstract section is fragile. Please re-write an abstract section, explain an obtained result and contribution, improve a proposed method, etc. Please delete unnecessary information 

Thank you for your comment. The abstract has been modified, taking your feedback into consideration.

2) This paper has more than spelling and grammatical errors. Please fix all of them.

We have checked and corrected the grammatical errors.

3) The authors should clearly state the limitations of the proposed method in other real applications.

Thank you for your pertinent comment, please refer to paragraph 4.3.4.

4) I suggest the authors add a table at the end of the literature review and compare the reviewed papers to clarify the research gap better.

Thank you for this pertinent comment, which has been added to the Related Work section.

5) The capitalization format of the title needs to be standardized. You can either capitalize the first letter of every word or use all lowercase letters.

For all titles, we proceeded to capitalize the first letter of every word.

6) Please re-write your contribution to this paper in the Introduction section.

Thanks for the relevant comment, please see the introduction section.

7) Please draw a flowchart of the proposed method.

Thank you, the architecture of our approach already exists; refer to Figure 3.

8) How did the authors set parameters for their proposed algorithm? Please make sensitivities of these parameters to the performance of their proposed algorithm!

Thank you for these interesting comment, the parameters of our proposed algorithm were chosen based on our own experience

9) Please make the Introduction and related work sections more productive using the following articles. Reading and using these articles and also cited in this article: A Multi-Objective Mutation-based Dynamic Harris Hawks Optimization for Botnet Detection in IoT, MOAEOSCA: an enhanced multi-objective hybrid artificial ecosystem-based optimization with sine cosine algorithm for feature selection in botnet detection in IoT, A Feature Selection based on the Farmland Fertility Algorithm for Improved Intrusion Detection Systems

Thank you for this pertinent comment, which has been added to the Related Work and references section. 

10) Expand the critical results in the conclusion. Focus on the main developments in the finale. Also, write the main contributions in the conclusion.

Thanks for the relevant comment, please see the Conclusion and Future Works section

11) Numerical results are good enough, but more explanations are required to analyze each figure presented.

Thanks for your comment, please see the Experimental Results and Discussion section

12) The simulation section is not detailed enough. Authors are suggested to provide more information about their employed data and simulation process.

Thank you for these interesting remarks, please see the Experimental Results and Discussion section.

13) Please change the title of the end section (Conclusions) to (Conclusions and Future Works) and write some future works.

Thank you for this pertinent comment, please see Conclusion and Future Works section.

14) All figures have low quality, and please improve all of them. 

Thank you for your comment, the quality of the figures has been improved.

Reviewer 2

CONSIDERATIONS

1) Why is shallow learning combined with feature extraction used to solve detection problems, rather than end-to-end deep learning? The author should explain this issue in depth from both theoretical and practical perspectives.

Thank you for your comment. Please refer to the end of the "Materials and Methods" section.

2) The specific structure of machine learning models should be introduced in detail. The reasons for adopting specific model structures and feature combinations should be explained.

Thank you for your comment, kindly see paragraph 4.3.2.

3) Many figures are blurry and should be re displayed.

Thank you for your comment, the quality of the figures has been improved.

4) Some deep learning methods should be considered for comparison.

Thank you for this pertinent comment; we compared our proposed model with LSTM because, according to the state of the art and based on our knowledge, it is widely used in the field of intrusion detection. We also conducted comparisons with previous studies that utilized other types of deep learning algorithms, such as DNN, RNN, and CNN.

5) Deep learning methods have achieved remarkable results in the field of feature extraction and learning. The following related work of deep learning techniques must be cited and discussed, including “Spectrum interference-based two-level data augmentation method in deep learning for automatic modulation classification,” Neural Computing & Applications, vol. 33, pp. 7723-7745, 2020. “MR-DCAE: Manifold regularization-based deep convolutional autoencoder for unauthorized broadcasting identification,” International Journal of Intelligent Systems, vol. 36, no. 12, pp. 7204-7238, 2021. “Fine-grained modulation classification using multi-scale radio transformer with dual-channel representation,” IEEE Communications Letters, vol. 26, no. 6, pp. 1298-1302, June. 2022. Application of wavelet-packet transform driven deep learning method in PM2. 5 concentration prediction: A case study of Qingdao, China. Sustainable Cities and Society, 2023, 92: 104486. “DL-PR: Generalized automatic modulation classification method based on deep learning with priori regularization,” Engineering Applications of Artificial Intelligence, 2023, 122: 106082.

Thank you for these interesting remarks, please see the introduction, where references have been added.

---

## [Decision Letter · Decision Letter 1]

30 Nov 2023

A Robust Intrusion Detection System based on a Shallow Learning Model and Feature Extraction Techniques

PONE-D-23-32323R1

Dear Dr. Douzi,

We’re pleased to inform you that your manuscript has been judged scientifically suitable for publication and will be formally accepted for publication once it meets all outstanding technical requirements.

Kind regards,

Mujeeb Ur Rehman, Ph.D.

Academic Editor

PLOS ONE

Additional Editor Comments (optional):

Reviewers' comments:

Reviewer's Responses to Questions

**Comments to the Author**

1. If the authors have adequately addressed your comments raised in a previous round of review and you feel that this manuscript is now acceptable for publication, you may indicate that here to bypass the “Comments to the Author” section, enter your conflict of interest statement in the “Confidential to Editor” section, and submit your "Accept" recommendation.

Reviewer #1: All comments have been addressed

Reviewer #2: All comments have been addressed

2. Is the manuscript technically sound, and do the data support the conclusions?

Reviewer #1: Yes

Reviewer #2: Yes

3. Has the statistical analysis been performed appropriately and rigorously? 

Reviewer #1: Yes

Reviewer #2: Yes

4. Have the authors made all data underlying the findings in their manuscript fully available?

Reviewer #1: Yes

Reviewer #2: Yes

5. Is the manuscript presented in an intelligible fashion and written in standard English?

Reviewer #1: Yes

Reviewer #2: Yes

6. Review Comments to the Author

Reviewer #1: The authors have completely addressed all my concerns, and I, therefore, recommend accepting this paper for publication.

Reviewer #2: This paper can be accepted for publication, since all the comments are well revised by the authors.

7. PLOS authors have the option to publish the peer review history of their article (what does this mean?). If published, this will include your full peer review and any attached files.

Reviewer #1: No

Reviewer #2: No

---

## [Editor Report · Acceptance letter]

13 Dec 2023

PONE-D-23-32323R1 

PLOS ONE

Dear Dr. Douzi, 

I'm pleased to inform you that your manuscript has been deemed suitable for publication in PLOS ONE. Congratulations! Your manuscript is now being handed over to our production team.

Kind regards, 

on behalf of

Dr. Mujeeb Ur Rehman 

Academic Editor

PLOS ONE